# LEARNING FACTORIZED MULTIMODAL REPRESENTATIONS

**Yao-Hung Hubert Tsai**[*1], **Paul Pu Liang**[*1],
**Amir Zadeh**[2], **Louis-Philippe Morency**[2], **Ruslan Salakhutdinov**[1]
{[1]Machine Learning Department, [2]Language Technologies Institute}, Carnegie Mellon University
{yaohungt,pliang,abagherz,morency,rsalakhu}@cs.cmu.edu

## ABSTRACT

Learning multimodal representations is a fundamentally complex research problem due to the presence of multiple heterogeneous sources of information. Although the presence of multiple modalities provides additional valuable information, there are two key challenges to address when learning from multimodal data: 1) models must learn the complex intra-modal and cross-modal interactions for prediction and 2) models must be robust to unexpected missing or noisy modalities during testing. In this paper, we propose to optimize for a joint generative-discriminative objective across multimodal data and labels. We introduce a model that factorizes representations into two sets of independent factors: *multimodal discriminative* and *modality-specific generative* factors. Multimodal discriminative factors are shared across all modalities and contain joint multimodal features required for discriminative tasks such as sentiment prediction. Modality-specific generative factors are unique for each modality and contain the information required for generating data. Experimental results show that our model is able to learn meaningful multimodal representations that achieve state-of-the-art or competitive performance on six multimodal datasets. Our model demonstrates flexible generative capabilities by conditioning on independent factors and can reconstruct missing modalities without significantly impacting performance. Lastly, we interpret our factorized representations to understand the interactions that influence multimodal learning.

## 1 INTRODUCTION

Multimodal machine learning involves learning from data across multiple modalities (Baltrušaitis et al., 2017). It is a challenging yet crucial research area with real-world applications in robotics (Liu et al., 2017), dialogue systems (Pittermann et al., 2010), intelligent tutoring systems (Petrovica et al., 2017), and healthcare diagnosis (Frantzidis et al., 2010). At the heart of many multimodal modeling tasks lies the challenge of learning rich representations from multiple modalities. For example, analyzing multimedia content requires learning multimodal representations across the language, visual, and acoustic modalities (Cho et al., 2015). Although the presence of multiple modalities provides additional valuable information, there are two key challenges to address when learning from multimodal data: 1) models must learn the complex intra-modal and cross-modal interactions for prediction (Zadeh et al., 2017), and 2) trained models must be robust to unexpected missing or noisy modalities during testing (Ngiam et al., 2011).

In this paper, we propose to optimize for a joint generative-discriminative objective across multimodal data and labels. The discriminative objective ensures that the representations learned are rich in intra-modal and cross-modal features useful towards predicting the label, while the generative objective allows the model to infer missing modalities at test time and deal with the presence of noisy modalities. To this end, we introduce the Multimodal Factorization Model (MFM in Figure 1) that factorizes multimodal representations into *multimodal discriminative* factors and *modality-specific generative* factors. Multimodal discriminative factors are shared across all modalities and contain joint multimodal features required for discriminative tasks. Modality-specific generative factors are unique for each modality and contain the information required for generating each modality. We believe that factorizing multimodal representations into different explanatory factors can help each factor focus on learning from a subset of the joint information across multimodal data and labels. This method is in contrast to jointly learning a single factor that summarizes all generative and

---

[*]equal contributions

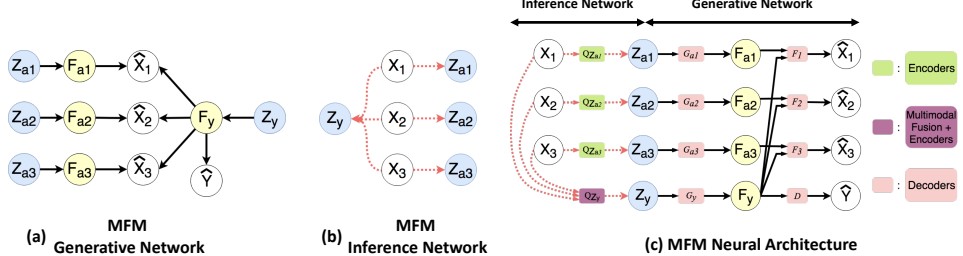

Figure 1: Illustration of the proposed Multimodal Factorization Model (MFM) with three modalities. MFM factorizes multimodal representations into *multimodal discriminative* factors $\mathbf{F_y}$ and *modality-specific generative* factors $\mathbf{F_{a\{1:M\}}}$. (a) MFM Generative Network with latent variables $\{\mathbf{Z_y}, \mathbf{Z_{a\{1:M\}}}\}$, factors $\{\mathbf{F_y}, \mathbf{F_{a\{1:M\}}}\}$, generated multimodal data $\hat{\mathbf{X}}_{1:3}$ and labels $\hat{\mathbf{Y}}$. (b) MFM Inference Network. (c) MFM Neural Architecture. Best viewed zoomed in and in color.

discriminative information (Srivastava & Salakhutdinov, 2012). To sum up, MFM defines a joint distribution over multimodal data, and by the conditional independence assumptions in the assumed graphical model, both generative and discriminative aspects are taken into account. Our model design further provides interpretability of the factorized representations.

Through an extensive set of experiments, we show that MFM learns improved multimodal representations with these characteristics: 1) The multimodal discriminative factors achieve state-of-the-art or competitive performance on six multimodal time series datasets. We also demonstrate that MFM can generalize by integrating it with other existing multimodal discriminative models. 2) MFM allows flexible generation concerning multimodal discriminative factors (labels) and modality-specific generative factors (styles). We further show that we can perform reconstruction of missing modalities from observed modalities without significantly impacting discriminative performance. Finally, we interpret our learned representations using information-based and gradient-based methods, allowing us to understand the contributions of individual factors towards multimodal prediction and generation.

## 2 MULTIMODAL FACTORIZATION MODEL

Multimodal Factorization Model (MFM) is a latent variable model (Figure 1(a)) with conditional independence assumptions over multimodal discriminative factors and modality-specific generative factors. According to these assumptions, we propose a factorization over the joint distribution of multimodal data (Section 2.1). Since exact posterior inference on this factorized distribution can be intractable, we propose an approximate inference algorithm based on minimizing a joint-distribution Wasserstein distance over multimodal data (Section 2.2). Finally, we derive the MFM objective by approximating the joint-distribution Wasserstein distance via a generalized mean-field assumption.

**Notation:** We define $\mathbf{X}_{1:M}$ as the multimodal data from $M$ modalities and $\mathbf{Y}$ as the labels, with joint distribution $P_{\mathbf{X}_{1:M}, \mathbf{Y}} = P(\mathbf{X}_{1:M}, \mathbf{Y})$. Let $\hat{\mathbf{X}}_{1:M}$ denote the generated multimodal data and $\hat{\mathbf{Y}}$ denote the generated labels, with joint distribution $P_{\hat{\mathbf{X}}_{1:M}, \hat{\mathbf{Y}}} = P(\hat{\mathbf{X}}_{1:M}, \hat{\mathbf{Y}})$.

### 2.1 FACTORIZED MULTIMODAL REPRESENTATIONS

To factorize multimodal representations into multimodal discriminative factors and modality-specific generative factors, MFM assumes a Bayesian network structure as shown in Figure 1(a). In this graphical model, factors $\mathbf{F_y}$ and $\mathbf{F_{a\{1:M\}}}$ are generated from mutually independent latent variables $\mathbf{Z} = [\mathbf{Z_y}, \mathbf{Z_{a\{1:M\}}}]$ with prior $P_\mathbf{Z}$. In particular, $\mathbf{Z_y}$ generates the multimodal discriminative factor $\mathbf{F_y}$ and $\mathbf{Z_{a\{1:M\}}}$ generate modality-specific generative factors $\mathbf{F_{a\{1:M\}}}$. By construction, $\mathbf{F_y}$ contributes to the generation of $\hat{\mathbf{Y}}$ while $\{\mathbf{F_y}, \mathbf{F_{a}}_i\}$ both contribute to the generation of $\hat{\mathbf{X}}_i$. As a result, the joint distribution $P(\hat{\mathbf{X}}_{1:M}, \hat{\mathbf{Y}})$ can be factorized as follows:

$$P(\hat{\mathbf{X}}_{1:M}, \hat{\mathbf{Y}}) = \int_{\mathbf{F}, \mathbf{Z}} P(\hat{\mathbf{X}}_{1:M}, \hat{\mathbf{Y}}|\mathbf{F})P(\mathbf{F}|\mathbf{Z})P(\mathbf{Z})d\mathbf{F}d\mathbf{Z}$$
$$= \int_{\substack{\mathbf{F_y}, \mathbf{F_{a\{1:M\}}} \\ \mathbf{Z_y}, \mathbf{Z_{a\{1:M\}}}}} \Big(P(\hat{\mathbf{Y}}|\mathbf{F_y})\prod_{i=1}^{M}P(\hat{\mathbf{X}}_i|\mathbf{F_{a}}_i, \mathbf{F_y})\Big)\Big(P(\mathbf{F_y}|\mathbf{Z_y})\prod_{i=1}^{M}P(\mathbf{F_{a}}_i|\mathbf{Z_{a}}_i)\Big)\Big(P(\mathbf{Z_y})\prod_{i=1}^{M}P(\mathbf{Z_{a}}_i)\Big)d\mathbf{F}d\mathbf{Z}, \quad (1)$$

with $d\mathbf{F} = d\mathbf{F_y}\prod_{i=1}^{M}d\mathbf{F_{a}}_i$ and $d\mathbf{Z} = d\mathbf{Z_y}\prod_{i=1}^{M}d\mathbf{Z_{a}}_i$.

Exact posterior inference in Equation 1 may be analytically intractable due to the integration over $\mathbf{Z}$. We therefore resort to using an approximate inference distribution $Q(\mathbf{Z}|\mathbf{X}_{1:M}, \mathbf{Y})$ as detailed in the

following subsection. As a result, MFM can be viewed as an autoencoding structure that consists of encoder (inference) and decoder (generative) modules (Figure 1(c)). The encoder module for $Q(\cdot|\cdot)$ allows us to easily sample $\mathbf{Z}$ from an approximate posterior. The decoder modules are parametrized according to the factorization of $P(\hat{\mathbf{X}}_{1:M}, \hat{\mathbf{Y}}|\mathbf{Z})$ as given by Equation 1 and Figure 1(a).

## 2.2 MINIMIZING JOINT-DISTRIBUTION WASSERSTEIN DISTANCE OVER MULTIMODAL DATA

Two common choices for approximate inference in autoencoding structures are Variational Autoencoders (VAEs) (Kingma & Welling, 2013) and Wasserstein Autoencoders (WAEs) (Zhao et al., 2017; Tolstikhin et al., 2017). The former optimizes the evidence lower bound objective (ELBO), and the latter derives an approximation for the primal form of the Wasserstein distance. We consider the latter since it simultaneously results in better latent factor disentanglement (Zhao et al., 2017; Rubenstein et al., 2018) and better sample generation quality than its counterparts (Chen et al., 2016; Higgins et al., 2016; Kingma & Welling, 2013). However, WAEs are designed for unimodal data and do not consider factorized distributions over latent variables that generate multimodal data. Therefore, we propose a variant for handling factorized joint distributions over multimodal data.

As suggested by Kingma & Welling (2013), we adopt the design of nonlinear mappings (i.e. neural network architectures) in the encoder and decoder (Figure 1 (c)). For the encoder $Q(\mathbf{Z}|\mathbf{X}_{1:M}, \mathbf{Y})$, we learn a deterministic mapping $Q_{enc} : \mathbf{X}_{1:M}, \mathbf{Y} \to \mathbf{Z}$ (Rubenstein et al., 2018; Tolstikhin et al., 2017). For the decoder, we define the generation process from latent variables as $G_y : \mathbf{Z}_\mathbf{y} \to \mathbf{F}_\mathbf{y}$, $G_{a\{1:M\}} :$ $\mathbf{Z}_{\mathbf{a}\{1:M\}} \to \mathbf{F}_{\mathbf{a}\{1:M\}}$, $D : \mathbf{F}_\mathbf{y} \to \hat{\mathbf{Y}}$, and $F_{1:M} : \mathbf{F}_\mathbf{y}, \mathbf{F}_{\mathbf{a}\{1:M\}} \to \hat{\mathbf{X}}_{1:M}$, where $G_y, G_{a\{1:M\}}, D$ and $F_{1:M}$ are deterministic functions parametrized by neural networks.

Let $W_c(P_{\mathbf{X}_{1:M}, \mathbf{Y}}, P_{\hat{\mathbf{X}}_{1:M}, \hat{\mathbf{Y}}})$ denote the joint-distribution Wasserstein distance over multimodal data under cost function $c_{Xi}$ and $c_Y$. We choose the squared cost $c(a,b) = \|a - b\|_2^2$, allowing us to minimize the 2-Wasserstein distance. The cost function can be defined not only on static data but also on time series data such as text, audio and videos. For example, given time series data $\mathbf{X} = [X^1, X^2, \cdots, X^T]$ and $\hat{\mathbf{X}} = [\hat{X}^1, \hat{X}^2, \cdots, \hat{X}^T]$, we define $c(\mathbf{X}, \hat{\mathbf{X}}) = \sum_{t=1}^{T} \|X^t - \hat{X}^t\|_2^2$.

With conditional independence assumptions in Equation 1, we express $W_c(P_{\mathbf{X}_{1:M}, \mathbf{Y}}, P_{\hat{\mathbf{X}}_{1:M}, \hat{\mathbf{Y}}})$ as:

**Proposition 1.** *For any functions* $G_y : \mathbf{Z}_\mathbf{y} \to \mathbf{F}_\mathbf{y}$, $G_{a\{1:M\}} : \mathbf{Z}_{\mathbf{a}\{1:M\}} \to \mathbf{F}_{\mathbf{a}\{1:M\}}$, $D : \mathbf{F}_\mathbf{y} \to \hat{\mathbf{Y}}$, *and* $F_{1:M} : \mathbf{F}_{\mathbf{a}\{1:M\}}, \mathbf{F}_\mathbf{y} \to \hat{\mathbf{X}}_{1:M}$, *we have* $W_c(P_{\mathbf{X}_{1:M}, \mathbf{Y}}, P_{\hat{\mathbf{X}}_{1:M}, \hat{\mathbf{Y}}}) =$

$$\inf_{Q_\mathbf{Z} = P_\mathbf{Z}} \mathbf{E}_{P_{\mathbf{X}_{1:M}, \mathbf{Y}}} \mathbf{E}_{Q(\mathbf{Z}|\mathbf{X}_{1:M}, \mathbf{Y})} \left[ \sum_{i=1}^{M} c_{X_i}\Big( \mathbf{X}_i, F_i\big(G_{ai}(\mathbf{Z}_{\mathbf{a}i}), G_y(\mathbf{Z}_\mathbf{y})\big)\Big) + c_Y\Big(\mathbf{Y}, D\big(G_y(\mathbf{Z}_\mathbf{y})\big)\Big) \right], \quad (2)$$

*where* $P_\mathbf{Z}$ *is the prior over* $\mathbf{Z} = [\mathbf{Z}_\mathbf{y}, \mathbf{Z}_{\mathbf{a}\{1,M\}}]$ *and* $Q_\mathbf{Z}$ *is the aggregated posterior of the proposed approximate inference distribution* $Q(\mathbf{Z}|\mathbf{X}_{1:M}, \mathbf{Y})$.

*Proof:* The proof is adapted from Tolstikhin *et al.* (Tolstikhin et al., 2017). The two differences are: (1) we show that $P(\hat{\mathbf{X}}_{1:M}, \hat{\mathbf{Y}}|\mathbf{Z} = z)$ are Dirac for all $z \in \mathcal{Z}$, and (2) we use the fact that $c((\mathbf{X}_{1:M}, \mathbf{Y}), (\hat{\mathbf{X}}_{1:M}, \hat{\mathbf{Y}})) = \sum_{i=1}^{M} c_{Xi}(\mathbf{X}_i, \hat{\mathbf{X}}_i) + c_Y(\mathbf{Y}, \hat{\mathbf{Y}})$. Please refer to the supplementary material for proof details. ∎

The constraint on $Q_\mathbf{Z} = P_\mathbf{Z}$ in Proposition 1 is hard to satisfy. To obtain a numerical solution, we first relax the constraint by performing a generalized mean field assumption on $Q$ according to the conditional independence as shown in the inference network of Figure 1 (b):

$$Q(\mathbf{Z}|\mathbf{X}_{1:M}, \mathbf{Y}) \coloneqq Q(\mathbf{Z}|\mathbf{X}_{1:M}) \coloneqq Q(\mathbf{Z}_\mathbf{y}|\mathbf{X}_{1:M}) \prod_{i=1}^{M} Q(\mathbf{Z}_{\mathbf{a}i}|\mathbf{X}_i). \quad (3)$$

The intuition here is based on our design that $\mathbf{Z}_\mathbf{y}$ generates the multimodal discriminative factor $\mathbf{F}_\mathbf{y}$ and $\mathbf{Z}_{\mathbf{a}\{1:M\}}$ generate modality-specific generative factors $\mathbf{F}_{\mathbf{a}\{1:M\}}$. Therefore, the inference for $\mathbf{Z}_\mathbf{y}$ should depend on all modalities $\mathbf{X}_{1:M}$ and the inference for $\mathbf{Z}_{\mathbf{a}i}$ should depend only on the specific modality $\mathbf{X}_i$. Following this assumption, we define $\mathcal{Q}$ as a nonparametric set of all encoders that fulfill the factorization in Equation 3. A penalty term is added into our objective to find the $Q(\mathbf{Z}|\cdot) \in \mathcal{Q}$ that is the closest to prior $P_\mathbf{Z}$, thereby approximately enforcing the constraint $Q_\mathbf{Z} = P_\mathbf{Z}$:

$$\min_{F, G_{a\{1:M\}}, G_y, D} \inf_{Q(\mathbf{Z}|\cdot) \in \mathcal{Q}} \mathbf{E}_{P_{\mathbf{X}_{1:M}, \mathbf{Y}}} \mathbf{E}_{Q(\mathbf{Z}_{\mathbf{a}1}|\mathbf{X}_1)} \cdots \mathbf{E}_{Q(\mathbf{Z}_{\mathbf{a}M}|\mathbf{X}_M)} \mathbf{E}_{Q(\mathbf{Z}_\mathbf{y}|\mathbf{X}_{1:M})}$$
$$\left[ \sum_{i=1}^{M} c_{X_i}\Big( \mathbf{X}_i, F\big(G_{ai}(\mathbf{Z}_{\mathbf{a}i}), G_y(\mathbf{Z}_\mathbf{y})\big)\Big) + c_Y\Big(\mathbf{Y}, D\big(G_y(\mathbf{Z}_\mathbf{y})\big)\Big) \right] + \lambda \mathcal{MMD}(Q_\mathbf{Z}, P_\mathbf{Z}), \quad (4)$$

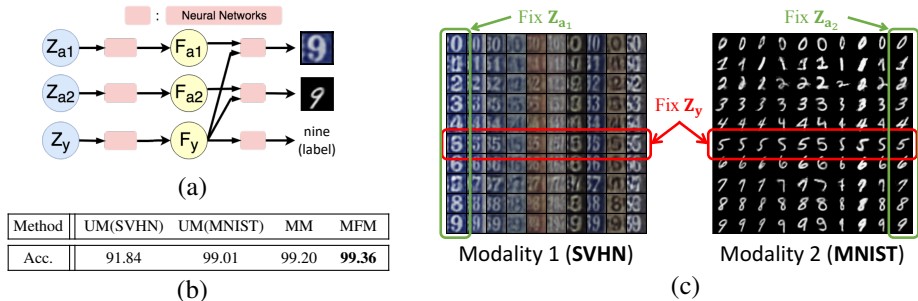

Figure 2: (a) MFM generative network for multimodal image dataset SVHN+MNIST, (b) unimodal and multimodal classification accuracies, and (c) conditional generation for SVHN and MNIST digits. MFM shows improved capabilities in digit prediction as well as flexible generation of both images based on labels and styles.

where $\lambda$ is a hyper-parameter and $\mathcal{MMD}$ is the Maximum Mean Discrepancy (Gretton et al., 2012) as a divergence measure between $Q_{\mathbf{Z}}$ and $P_{\mathbf{Z}}$. The prior $P_{\mathbf{Z}}$ is chosen as a centered isotropic Gaussian $\mathcal{N}(\mathbf{0}, \mathbf{I})$, so that it implicitly enforces independence between the latent variables $\mathbf{Z} = [\mathbf{Z}_{\mathbf{y}}, \mathbf{Z}_{\mathbf{a}\{1,M\}}]$ (Higgins et al., 2016; Kingma & Welling, 2013; Rubenstein et al., 2018).

Equation 4 represents our hybrid generative-discriminative optimization objective over multimodal data: the first loss term $\sum_{i=1}^{M} c_{X_i}(\mathbf{X}_i, F(G_{ai}(\mathbf{Z}_{\mathbf{a}i}), G_y(\mathbf{Z}_{\mathbf{y}})))$ is the generative objective based on reconstruction of multimodal data and the second term $c_Y(\mathbf{Y}, D(G_y(\mathbf{Z}_{\mathbf{y}})))$ is the discriminative objective. In practice we compute the expectations in Equation 4 using empirical estimates over the training data. The neural architecture of MFM is illustrated in Figure 1(c).

## 2.3 SURROGATE INFERENCE FOR MISSING MODALITIES

A key challenge in multimodal learning involves dealing with missing modalities. A good multimodal model should be able to infer the missing modality conditioned on the observed modalities and perform predictions based only on the observed modalities. To achieve this objective, the inference process of MFM can be easily adapted using a surrogate inference network to reconstruct the missing modality given the observed modalities. Formally, let $\Phi$ denote the surrogate inference network. The generation of missing modality $\hat{\mathbf{X}}_1$ given the observed modalities $\mathbf{X}_{2:M}$ can be formulated as

$$\Phi^* = \underset{\Phi}{\operatorname{argmin}} \, \mathbf{E}_{P_{\mathbf{X}_{2:M}, \hat{\mathbf{x}}_1}} \left( -\log P_{\Phi}(\hat{\mathbf{X}}_1 | \mathbf{X}_{2:M}) \right)$$

$$\text{with } P_{\Phi}(\hat{\mathbf{X}}_1 | \mathbf{X}_{2:M}) \coloneqq \int P(\hat{\mathbf{X}}_1 | \mathbf{Z}_{\mathbf{a}1}, \mathbf{Z}_{\mathbf{y}}) Q_{\Phi}(\mathbf{Z}_{\mathbf{a}1} | \mathbf{X}_{2:M}) Q_{\Phi}(\mathbf{Z}_{\mathbf{y}} | \mathbf{X}_{2:M}) d\mathbf{Z}_{\mathbf{a}1} d\mathbf{Z}_{\mathbf{y}}. \tag{5}$$

Similar to Section 2.2, we use deterministic mappings in $Q_{\Phi}(\cdot|\cdot)$ and $Q_{\Phi}(\mathbf{Z}_{\mathbf{y}}|\cdot)$ is also used for prediction $P_{\Phi}(\hat{\mathbf{Y}}|\mathbf{X}_{2:M}) \coloneqq \int P(\hat{\mathbf{Y}}|\mathbf{Z}_{\mathbf{y}}) Q_{\Phi}(\mathbf{Z}_{\mathbf{y}}|\mathbf{X}_{2:M}) d\mathbf{Z}_{\mathbf{y}}$. Equation 5 suggests that in the presence of missing modalities, we only need to infer the latent codes rather than the entire modality.

## 2.4 ENCODER AND DECODER DESIGN

We now discuss the implementation choices for the MFM neural architecture in Figure 1(c). The encoder $Q(\mathbf{Z}_{\mathbf{y}}|\mathbf{X}_{1:M})$ can be parametrized by any model that performs multimodal fusion (Morency et al., 2011; Zadeh et al., 2017). For multimodal image datasets, we adopt Convolutional Neural Networks (CNNs) and Fully-Connected Neural Networks (FCNNs) with late fusion (Nojavanasghari et al., 2016) as our encoder $Q(\mathbf{Z}_{\mathbf{y}}|\mathbf{X}_{1:M})$. The remaining functions in MFM are also parametrized by CNNs and FCNNs. For multimodal time series datasets, we choose the Memory Fusion Network (MFN) (Zadeh et al., 2018a) as our multimodal encoder $Q(\mathbf{Z}_{\mathbf{y}}|\mathbf{X}_{1:M})$. We use Long Short-term Memory (LSTM) networks (Hochreiter & Schmidhuber, 1997) for functions $Q(\mathbf{Z}_{\mathbf{a}\{1:M\}}|\mathbf{X}_{1:M})$, decoder LSTM networks (Cho et al., 2014) for functions $F_{1:M}$, and FCNNs for functions $G_y$, $G_{a\{1:M\}}$ and $D$. Details are provided in the appendix and the code is available at `<anonymous>`.

## 3 EXPERIMENTS

In order to show that MFM learns multimodal representations that are discriminative, generative and interpretable, we design the following experiments. We begin with a multimodal synthetic image dataset that allows us to examine whether MFM displays discriminative and generative capabilities from factorized latent variables. Utilizing image datasets allows us to clearly visualize the generative capabilities of MFM. We then transition to six more challenging real-world multimodal video datasets

Table 1: Results for multimodal speaker traits recognition on POM, multimodal sentiment analysis on CMU-MOSI, ICT-MMMO, YouTube, MOUD, and multimodal emotion recognition on IEMOCAP. SOTA1 and SOTA2 refer to the previous best and second best state-of-the-art respectively, and $\Delta_{SOTA}$ shows improvement over SOTA1. Symbols depict the baseline giving the result: $\#$ *MFN*, $\ddagger$ *MARN*, $\star$ *TFN*, $\dagger$ *BC-LSTM*, $\diamond$ *MV-LSTM*, $\S$ *EF-LSTM*, $\flat$ *DF*, $\heartsuit$ *SVM*, $\bullet$ *RF*. For detailed tables with results for all models, please refer to the appendix.

| Dataset | POM Personality Traits | | | | | | | | | | | | | | | |
|---|---|---|---|---|---|---|---|---|---|---|---|---|---|---|---|---|
| Task | Con | Pas | Voi | Dom | Cre | Viv | Exp | Ent | Res | Tru | Rel | Out | Tho | Ner | Per | Hum |
| Metric | | | | | | | | $r$ | | | | | | | | |
| SOTA2 | $0.359^\dagger$ | $0.425^\ddagger$ | $0.166^\ddagger$ | $0.235^\ddagger$ | $0.358^\ddagger$ | $0.417^\dagger$ | $0.450^\dagger$ | $0.378^\ddagger$ | $0.295^\diamond$ | $0.237^\diamond$ | $0.215^\ddagger$ | $0.238^\diamond$ | $0.363^\dagger$ | $0.258^\diamond$ | $0.344^\dagger$ | $0.319^\dagger$ |
| SOTA1 | $0.395^\#$ | $0.428^\#$ | $0.193^\#$ | $0.313^\#$ | $0.367^\#$ | $0.431^\#$ | $0.452^\#$ | $0.395^\#$ | $0.333^\#$ | $\mathbf{0.296}^\#$ | $0.255^\#$ | $0.259^\#$ | $0.381^\#$ | $0.318^\#$ | $\mathbf{0.377}^\dagger$ | $0.386^\#$ |
| MFM | **0.431** | **0.450** | **0.197** | **0.411** | **0.380** | **0.448** | **0.467** | **0.452** | **0.368** | 0.212 | **0.309** | **0.333** | **0.404** | **0.333** | 0.334 | **0.408** |
| $\Delta_{SOTA}$ | ↑0.036 | ↑0.022 | ↑0.004 | ↑0.097 | ↑0.013 | ↑0.017 | ↑0.015 | ↑0.057 | ↑0.035 | – | ↑0.054 | ↑0.074 | ↑0.023 | ↑0.015 | – | ↑0.022 |

| Dataset | CMU-MOSI | | | | | ICT-MMMO | | YouTube | | MOUD | |
|---|---|---|---|---|---|---|---|---|---|---|---|
| Task | Sentiment | | | | | Sentiment | | Sentiment | | Sentiment | |
| Metric | Acc_7 | Acc_2 | F1 | MAE | $r$ | Acc_2 | F1 | Acc_3 | F1 | Acc_2 | F1 |
| SOTA2 | $34.1^\#$ | $77.1^\ddagger$ | $77.0^\ddagger$ | $0.968^\ddagger$ | $0.625^\ddagger$ | $72.5^\star$ | $72.6^\star$ | $48.3^\dagger$ | $45.1^\dagger$ | $81.1^\#$ | $80.4^\#$ |
| SOTA1 | $34.7^\ddagger$ | $77.4^\ddagger$ | $77.3^\ddagger$ | $0.965^\#$ | $0.632^\#$ | $73.8^\#$ | $73.1^\#$ | $51.7^\#$ | $51.6^\#$ | $81.1^\ddagger$ | $81.2^\ddagger$ |
| MFM | **36.2** | **78.1** | **78.1** | **0.951** | **0.662** | **81.3** | **79.2** | **53.3** | **52.4** | **82.1** | **81.7** |
| $\Delta_{SOTA}$ | ↑1.5 | ↑0.7 | ↑0.8 | ↓0.014 | ↑0.030 | ↑7.5 | ↑6.1 | ↑1.6 | ↑0.8 | ↑1.0 | ↑0.5 |

| Dataset | IEMOCAP Emotions | | | | | | | | | | | |
|---|---|---|---|---|---|---|---|---|---|---|---|---|
| Task | Happy | | Sad | | Angry | | Frustrated | | Excited | | Neutral | |
| Metric | Acc_2 | F1 | Acc_2 | F1 | Acc_2 | F1 | Acc_2 | F1 | Acc_2 | F1 | Acc_2 | F1 |
| SOTA2 | $86.7^\ddagger$ | $84.2^\S$ | $83.4^\star$ | $81.7^\dagger$ | $85.1^\diamond$ | $84.5^\S$ | $79.5^\ddagger$ | $76.6^\ddagger$ | $89.6^\ddagger$ | $86.3^\#$ | $68.8^\S$ | $67.1^\S$ |
| SOTA1 | $90.1^\#$ | $85.3^\#$ | $85.8^\#$ | $82.8^\star$ | $87.0^\#$ | $86.0^\#$ | $80.3^\#$ | $\mathbf{76.8}^\ddagger$ | $89.8^\#$ | $87.1^\ddagger$ | $71.8^\#$ | $\mathbf{68.5}^\S$ |
| MFM | **90.2** | **85.8** | **88.4** | **86.1** | **87.5** | **86.7** | **80.4** | 74.5 | **90.0** | 87.1 | **72.1** | 68.1 |
| $\Delta_{SOTA}$ | ↑0.1 | ↑0.5 | ↑2.6 | ↑3.3 | ↑0.5 | ↑0.7 | ↑0.1 | – | ↑0.2 | – | ↑0.3 | – |

to 1) rigorously evaluate the discriminative capabilities of MFM in comparison with existing baselines, 2) analyze the importance of each design component through ablation studies, 3) assess the robustness of MFM's modality reconstruction and prediction capabilities to missing modalities, and 4) interpret the learned representations using information-based and gradient-based methods to understand the contributions of individual factors towards multimodal prediction and generation.

## 3.1 MULTIMODAL SYNTHETIC IMAGE DATASET

In this section, we study MFM on a synthetic image dataset that considers SVHN (Netzer et al., 2011) and MNIST (Lecun et al., 1998) as the two modalities. SVHN and MNIST are images with different styles but the same labels (digits 0 ∼ 9). We randomly pair $100,000$ SVHN and MNIST images that have the same label, creating a multimodal dataset which we call SVHN+MNIST. $80,000$ pairs are used for training and the rest for testing. To justify that MFM is able to learn improved multimodal representations, we show both classification and generation results on SVHN+MNIST in Figure 2.

**Prediction:** We perform experiments on both unimodal and multimodal classification tasks. UM denotes a unimodal baseline that performs prediction given only one modality as input and MM denotes a multimodal discriminative baseline that performs prediction given both images (Noja-vanasghari et al., 2016). We compare the results for UM(SVHN), UM(MNIST), MM and MFM on SVHN+MNIST in Figure 2(b). We achieve better classification performance from unimodal to multimodal which is not surprising since more information is given. More importantly, MFM outperforms MM, which suggests that MFM learns improved factorized representations for discriminative tasks.

**Generation:** We generate images using the MFM generative network (Figure 2(a)). We fix one variable out of $\mathbf{Z} = [\mathbf{Z_{a1}}, \mathbf{Z_{a2}},$ and $\mathbf{Z_y}]$ and randomly sample the other two variables from prior $P_\mathbf{Z}$. From Figure 2(c), we observe that MFM shows flexible generation of SVHN and MNIST images based on labels and styles. This suggests that MFM is able to factorize multimodal representations into multimodal discriminative factors (labels) and modality-specific generative factors (styles).

## 3.2 MULTIMODAL TIME SERIES DATASETS

In this section, we transition to more challenging multimodal time series datasets. All the datasets consist of monologue videos. Features are extracted from the language (GloVe word embeddings (Pen-

| Model | Multimodal Disc. Factor | Hybrid Gen.-Disc. Objective | Factorized Gen.-Disc. Factors | Mod.-Spec. Gen. Factors | CMU-MOSI | | | | | | | |
|---|---|---|---|---|---|---|---|---|---|---|---|---|
| | | | | | $\hat{\mathbf{X}}$. Reconstruction | | | $\hat{\mathbf{Y}}$ Prediction | | | | |
| | | | | | MSE $(\ell)$ | MSE $(a)$ | MSE $(v)$ | Acc_7 | Acc_2 | F1 | MAE | $r$ |
| $\mathbf{M_A}$ | no | no | - | - | - | - | - | 33.2 | 75.2 | 75.2 | 1.020 | 0.616 |
| $\mathbf{M_B}$ | yes | no | - | - | - | - | - | 34.1 | 77.4 | 77.3 | 0.965 | 0.632 |
| $\mathbf{M_C}$ | no | yes | no | - | 0.0413 | 0.0509 | 0.0220 | 34.8 | 75.9 | 76.0 | 0.979 | 0.640 |
| $\mathbf{M_D}$ | yes | yes | no | - | 0.0413 | 0.0486 | 0.0223 | 35.0 | 77.4 | 77.2 | 0.960 | 0.649 |
| $\mathbf{M_E}$ | yes | yes | yes | no | 0.0397 | 0.0452 | 0.0211 | 35.9 | 77.3 | 77.2 | 0.956 | 0.661 |
| MFM | yes | yes | yes | yes | **0.0391** | **0.0384** | **0.0183** | **36.2** | **78.1** | **78.1** | **0.951** | **0.662** |

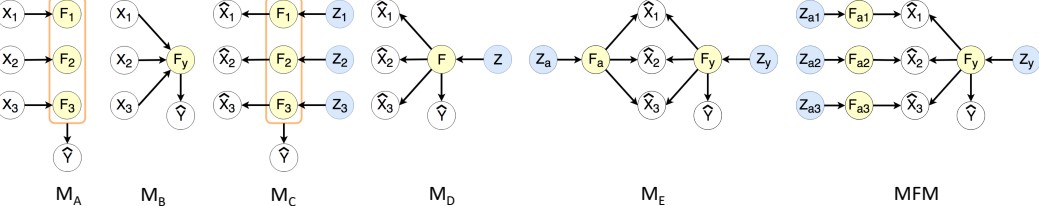

Figure 3: Models used in the ablation studies of MFM. Each model removes a design component from our model. Modality reconstruction and sentiment prediction results are reported on CMU-MOSI with best results in bold. Factorizing multimodal representations into multimodal discriminative factors and modality-specific generative factors are crucial for improved performance.

nington et al., 2014)), visual (Facet (iMotions, 2017)), and acoustic (COVAREP (Degottex et al., 2014)) modalities. For a detailed description of feature extraction, please refer to the appendix.

We consider the following six datasets across three domains: 1) Multimodal Personality Trait Recognition: **POM** (Park et al., 2014) contains 903 movie review videos annotated for the following personality traits: confident (con), passionate (pas), voice pleasant (voi), dominant (dom), credible (cre), vivid (viv), expertise (exp), entertaining (ent), reserved (res), trusting (tru), relaxed (rel), outgoing (out), thorough (tho), nervous (ner), persuasive (per) and humorous (hum). The short form is indicated in parenthesis. 2) Multimodal Sentiment Analysis: **CMU-MOSI** (Zadeh et al., 2016) is a collection of 2199 monologue opinion video clips annotated with sentiment. **ICT-MMMO** (Wöllmer et al., 2013) consists of 340 online social review videos annotated for sentiment. **YouTube** (Morency et al., 2011) contains 269 product review and opinion video segments from YouTube each annotated for sentiment. **MOUD** (Perez-Rosas et al., 2013) consists of 79 product review videos in Spanish. Each video consists of multiple segments labeled as either positive, negative or neutral sentiment. 3) Multimodal Emotion Recognition: **IEMOCAP** (Busso et al., 2008) consists of 302 videos of recorded dyadic dialogues. The videos are divided into multiple segments each annotated for the presence of 6 discrete emotions (happy, sad, angry, frustrated, excited and neutral), resulting in a total of 7318 segments in the dataset. We report results using the following metrics: Acc_$C$ = multiclass accuracy across $C$ classes, F1 = F1 score, MAE = Mean Absolute Error, $r$ = Pearson's correlation.

**Prediction:** We first compare the performance of MFM with existing multimodal prediction methods. For a detailed description of the baselines, please refer to the appendix. From Table 1, we first observe that the best performing baseline results are achieved by different models across different datasets (most notably MFN, MARN, and TFN). On the other hand, MFM consistently achieves state-of-the-art or competitive results for all six multimodal datasets. We believe that the multimodal discriminative factor $\mathbf{F_y}$ in MFM has successfully learned more meaningful representations by distilling discriminative features. This highlights the benefit of learning factorized multimodal representations towards discriminative tasks. Furthermore, MFM is *model-agnostic* and can be applied to other multimodal encoders $Q(\mathbf{Z_y}|\mathbf{X}_{1:M})$. We perform experiments to show consistent improvements in discriminative performance for several choices of the encoder: EF-LSTM (Morency et al., 2011) and TFN (Zadeh et al., 2017). For Acc_2 on CMU-MOSI, our factorization framework improves the performance of EF-LSTM from 74.3 to **75.2** and TFN from 74.6 to **75.5**.

**Ablation Study:** In Figure 3, we present the models $\mathbf{M}_{\{\mathbf{A},\mathbf{B},\mathbf{C},\mathbf{D},\mathbf{E}\}}$ used for ablation studies. These models are designed to analyze the effects of using a multimodal discriminative factor, a hybrid generative-discriminative objective, factorized generative-discriminative factors and modality-specific generative factors towards both modality reconstruction and label prediction. The simplest variant is $\mathbf{M_A}$ which represents a purely discriminative model without a joint multimodal discriminative factor (i.e. early fusion (Morency et al., 2011)). $\mathbf{M_B}$ models a joint multimodal discriminative factor which incorporates more general multimodal fusion encoders (Zadeh et al., 2018a). $\mathbf{M_C}$ extends

Table 2: The effect of missing modalities on multimodal data reconstruction and sentiment prediction on CMU-MOSI. MFM with surrogate inference is able to better handle missing modalities during test time as compared to the purely generative (Seq2Seq) or purely discriminative baselines.

| Task | $\hat{\mathbf{X}}$. Reconstruction | | | $\hat{\mathbf{Y}}$ Prediction | | | | |
|---|---|---|---|---|---|---|---|---|
| Metric | MSE ($\ell$) | MSE ($a$) | MSE ($v$) | Acc_7 | Acc_2 | F1 | MAE | $r$ |
| Purely Generative and Discriminative Baselines | | | | | | | | |
| $\ell$(anguage) missing | 0.0411 | - | - | 19.4 | 59.6 | 59.7 | 1.386 | 0.225 |
| $a$(udio) missing | - | 0.0533 | - | 34.0 | 73.5 | 73.4 | 1.024 | 0.615 |
| $v$(isual) missing | - | - | 0.0220 | 33.7 | 75.4 | 75.4 | 0.996 | 0.634 |
| Multimodal Factorization Model (MFM) | | | | | | | | |
| $\ell$(anguage) missing | 0.0403 | - | - | 21.7 | 62.0 | 61.7 | 1.313 | 0.236 |
| $a$(udio) missing | - | 0.0468 | - | 35.4 | 74.3 | 74.3 | 1.011 | 0.603 |
| $v$(isual) missing | - | - | 0.0215 | 35.0 | 76.4 | 76.3 | 0.990 | 0.635 |
| all present | **0.0391** | **0.0384** | **0.0182** | **36.2** | **78.1** | **78.1** | **0.951** | **0.662** |

$\mathbf{M_A}$ by optimizing a hybrid generative-discriminative objective over modality-specific factors. $\mathbf{M_D}$ extends $\mathbf{M_B}$ by optimizing a hybrid generative-discriminative objective over a joint multimodal factor (resembling Srivastava & Salakhutdinov (2012)). $\mathbf{M_E}$ factorizes the representation into separate generative and discriminative factors. Finally, MFM is obtained from $\mathbf{M_E}$ by using modality-specific generative factors instead of a joint multimodal generative factor.

From the table in Figure 3, we observe the following general trends. For sentiment prediction, using 1) a multimodal discriminative factor outperforms modality-specific discriminative factors ($\mathbf{M_D} > \mathbf{M_C}$, $\mathbf{M_B} > \mathbf{M_A}$), and 2) adding generative capabilities to the model improves performance ($\mathbf{M_C} > \mathbf{M_A}$, $\mathbf{M_E} > \mathbf{M_B}$). For both sentiment prediction and modality reconstruction, 3) factorizing into separate generative and discriminative factors improves performance ($\mathbf{M_E} > \mathbf{M_D}$), and 4) using modality-specific generative factors outperforms multimodal generative factors (MFM > $\mathbf{M_E}$). These observations support our design decisions of factorizing multimodal representations into multimodal discriminative factors and modality-specific generative factors.

**Missing Modalities:** We now evaluate the performance of MFM in the presence of missing modalities using the surrogate inference model as described in Subsection 2.3. We compare with two baselines: 1) a purely generative Seq2Seq model (Cho et al., 2014) $\Phi_G$ from observed modalities to missing modalities by optimizing $\mathbf{E}_{P_{\mathbf{X}_{1:M}}} \left( -\log P_{\Phi_D}(\mathbf{X}_1 | \mathbf{X}_{2:M}) \right)$, and 2) a purely discriminative model $\Phi_D$ from observed modalities to the label by optimizing $\mathbf{E}_{P_{\mathbf{X}_{2:M}, \mathbf{Y}}} \left( -\log P_{\Phi_G}(\mathbf{Y} | \mathbf{X}_{2:M}) \right)$. Both models are modified from MFM by using only the two observed modalities as input and not explicitly accounting for missing modalities. We compare the reconstruction error of each modality (language, visual and acoustic) as well as the performance on sentiment prediction.

Table 2 shows that MFM with missing modalities outperforms the generative ($\Phi_G$) or discriminative baselines ($\Phi_D$) in terms of modality reconstruction and sentiment prediction. Additionally, MFM with missing modalities performs close to MFM with all modalities observed. This fact indicates that MFM can learn representations that are relatively robust to missing modalities. In addition, discriminative performance is most affected when the language modality is missing, which is consistent with prior work which indicates that language is most informative in human multimodal language (Zadeh et al., 2017). On the other hand, sentiment prediction is more robust to missing acoustic and visual features. Finally, we observe that reconstructing the low-level acoustic and visual features is easier as compared to the high-dimensional language features that contain high-level semantic meaning.

**Interpretation of Multimodal Representations:** We devise two methods to study how individual factors in MFM influence the dynamics of multimodal prediction and generation. These interpretation methods represent both overall trends and fine-grained analysis that could be useful towards deeper understandings of multimodal representation learning. For more details, please refer to the appendix.

Firstly, an information-based interpretation method is chosen to summarize the contribution of each modality towards the multimodal representations. Since $\mathbf{F_y}$ is a common cause of $\hat{\mathbf{X}}_{1:M}$, we can compare $\mathrm{MI}(\mathbf{F_y}, \hat{\mathbf{X}}_1), \cdots, \mathrm{MI}(\mathbf{F_y}, \hat{\mathbf{X}}_M)$, where $\mathrm{MI}(\cdot, \cdot)$ denotes the mutual information measure between $\mathbf{F_y}$ and generated modality $\hat{\mathbf{X}}_i$. Higher $\mathrm{MI}(\mathbf{F_y}, \hat{\mathbf{X}}_i)$ indicates greater contribution from $\mathbf{F_y}$ to $\hat{\mathbf{X}}_i$. Figure 4 reports the ratios $r_i = \mathrm{MI}(\mathbf{F_y}, \hat{\mathbf{X}}_i)/\mathrm{MI}(\mathbf{F_{a}}_i, \hat{\mathbf{X}}_i)$ which measure a normalized version of the mutual information between $\mathbf{F_{a}}_i$ and $\hat{\mathbf{X}}_i$. We observe that on CMU-MOSI, the language modality is most informative towards sentiment prediction, followed by the acoustic modality. We

| Ratio | $r_\ell$ | $r_v$ | $r_a$ |
|---|---|---|---|
| CMU-MOSI | 0.307 | 0.030 | 0.107 |

Figure 4: Analyzing the multimodal representations learnt in MFM via information-based (entire dataset) and gradient-based interpretation methods (single video) on CMU-MOSI.

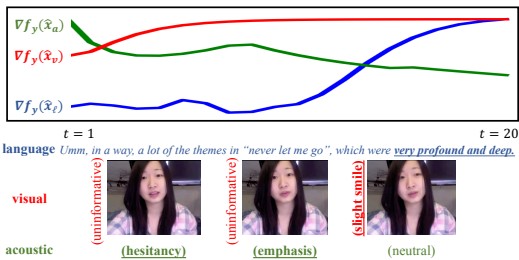

believe that this result represents a prior over the expression of sentiment in human multimodal language and is closely related to the connections between language and speech (Kuhl, 2000).

Secondly, a gradient-based interpretation method to used analyze the contribution of each modality for every time step in multimodal time series data. We measure the gradient of the generated modality with respect to the target factors (e.g., $\mathbf{F_y}$). Let $\{x_1, x_2, \cdots, x_M\}$ denote multimodal time series data where $x_i$ represents modality $i$, and $\hat{x}_i = [\hat{x}_i^1, \cdots, \hat{x}_i^t, \cdots, \hat{x}_i^T]$ denote generated modality $i$ across time steps $t \in [1, T]$. The gradient $\nabla_{f_y}(\hat{x}_i)$ measures the extent to which changes in factor $f_y \sim P(\mathbf{F_y}|\mathbf{X}_{1:M} = x_{1:M})$ influences the generation of sequence $\hat{x}_i$. Figure 4 plots $\nabla_{f_y}(\hat{x}_i)$ for a video in CMU-MOSI. We observe that multimodal communicative behaviors that are indicative of speaker sentiment such as positive words (e.g. "very profound and deep") and informative acoustic features (e.g. hesitant and emphasized tone of voice) indeed correspond to increases in $\nabla_{f_y}(\hat{x}_i)$.

## 4 RELATED WORK

The two main pillars of research in multimodal representation learning have considered the discriminative and generative objectives individually. Discriminative representation learning (Liang et al., 2018; Chen et al., 2017; Chaplot et al., 2017; Frome et al., 2013; Socher et al., 2013; Tsai et al., 2017) models the conditional distribution $P(\mathbf{Y}|\mathbf{X}_{1:M})$. Since these approaches are not concerned with modeling $P(\mathbf{X}_{1:M})$ explicitly, they use parameters more efficiently to model $P(\mathbf{Y}|\mathbf{X}_{1:M})$. For instance, recent works learn visual representations that are maximally dependent with linguistic attributes for improving one-shot image recognition (Tsai & Salakhutdinov, 2017) or introduce tensor product mechanisms to model interactions between the language, visual and acoustic modalities (Liu et al., 2018; Zadeh et al., 2017). On the other hand, generative representation learning captures the interactions between modalities by modeling the joint distribution $P(\mathbf{X}_1, \cdots, \mathbf{X}_M)$ using either undirected graphical models (Srivastava & Salakhutdinov, 2012), directed graphical models (Suzuki et al., 2016b), or neural networks (Sohn et al., 2014). Some generative approaches compress multimodal data into lower-dimensional feature vectors which can be used for discriminative tasks (Pham et al., 2018; Ngiam et al., 2011). To unify the advantages of both approaches, MFM factorizes multimodal representations into generative and discriminative components and optimizes for a joint objective.

Factorized representation learning resembles learning disentangled data representations which have been shown to improve the performance on many tasks (Kulkarni et al., 2015; Lake et al., 2017; Higgins et al., 2016; Bengio et al., 2013). Several methods involve specifying a fixed set of latent attributes that individually control particular variations of data and performing supervised training (Cheung et al., 2014; Karaletsos et al., 2015; Yang et al., 2015; Reed et al., 2014; Zhu et al., 2014), assuming an isotropic Gaussian prior over latent variables to learn disentangled generative representations (Kingma & Welling, 2013; Rubenstein et al., 2018) and learning latent variables in charge of specific variations in the data by maximizing the mutual information between a subset of latent variables and the data (Chen et al., 2016). However, these methods study factorization of a single modality. MFM factorizes multimodal representations and demonstrates the importance of modality-specific and multimodal factors towards generation and prediction. A concurrent and parallel work that factorizes latent factors in multimodal data was proposed by Hsu & Glass (2018). They differ from us in the graphical model design, discriminative objective, prior matching criterion, and scale of experiments. We provide a detailed comparison with their model in the appendix.

## 5 CONCLUSION

In this paper, we proposed the Multimodal Factorization Model (MFM) for multimodal representation learning. MFM factorizes the multimodal representations into two sets of independent factors: *multimodal discriminative* factors and *modality-specific generative* factors. The multimodal discriminative

factor achieves state-of-the-art or competitive results on six multimodal datasets. The modality-specific generative factors allow us to generate data based on factorized variables, account for missing modalities, and have a deeper understanding of the interactions involved in multimodal learning. Our future work will explore extensions of MFM for video generation, semi-supervised learning, and unsupervised learning. We believe that MFM sheds light on the advantages of learning factorizing multimodal representations and potentially opens up new horizons for multimodal machine learning.

## ACKNOWLEDGEMENTS

This work was supported in part by the DARPA grants D17AP00001 and FA875018C0150, Office of Naval Research, Apple, and Google focused award. We would also like to acknowledge NVIDIA's GPU support. This material is also based upon work partially supported by the National Science Foundation (Award 1750439). Any opinions, findings, and conclusions or recommendations expressed in this material are those of the author(s) and do not necessarily reflect the views of National Science Foundation, and no official endorsement should be inferred.

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

## A  PROOF OF PROPOSITION 1

To simplify the proof, we first prove it for the unimodal case by considering the Wasserstein distance between $P_{\mathbf{X},\mathbf{Y}}$ and $P_{\hat{\mathbf{X}},\hat{\mathbf{Y}}}$.

### A.1  UNIMODAL JOINT-DISTRIBUTION WASSERSTEIN DISTANCE

**Proposition 2.** *For any functions* $G_y : \mathbf{Z_y} \to \mathbf{F_y}$, $G_a : \mathbf{Z_a} \to \mathbf{F_a}$, $D : \mathbf{F_y} \to \hat{\mathbf{Y}}$, *and* $F : \mathbf{F_a}, \mathbf{F_y} \to \hat{\mathbf{X}}$, *we have*

$$W_c(P_{\mathbf{X},\mathbf{Y}}, P_{\hat{\mathbf{X}},\hat{\mathbf{Y}}}) = \inf_{Q_{\mathbf{Z}} = P_{\mathbf{Z}}} \mathbf{E}_{P_{\mathbf{X},\mathbf{Y}}} \mathbf{E}_{Q(\mathbf{Z}|\mathbf{X})} \left[ c_X\Big(\mathbf{X}, F\big(G_a(\mathbf{Z_a}), G_y(\mathbf{Z_y})\big)\Big) + c_Y\Big(\mathbf{Y}, D\big(G_y(\mathbf{Z_y})\big)\Big) \right],$$
(6)

*where* $W_c$ *is the Wasserstein distance under cost function* $c_X$ *and* $c_Y$, $P_{\mathbf{Z}}$ *is the prior over* $\mathbf{Z} = [\mathbf{Z_a}, \mathbf{Z_y}]$ *and* $Q_{\mathbf{Z}}$ *is the aggregated posterior of the proposed inference distribution* $Q(\mathbf{Z}|\mathbf{X})$.

*Proof:* See the following.

To begin the proof, we abuse some notations as follows.

By definition, the Wasserstein distance under cost function $c$ between $P_{\mathbf{X},\mathbf{Y}}$ and $P_{\hat{\mathbf{X}},\hat{\mathbf{Y}}}$ is

$$W_c(P_{\mathbf{X},\mathbf{Y}}, P_{\hat{\mathbf{X}},\hat{\mathbf{Y}}}) := \inf_{\Gamma \in \mathcal{P}\left((\mathbf{X},\mathbf{Y}) \sim P_{\mathbf{X},\mathbf{Y}}, (\hat{\mathbf{X}},\hat{\mathbf{Y}}) \sim P_{\hat{\mathbf{X}},\hat{\mathbf{Y}}}\right)} \mathbf{E}_{\left((\mathbf{X},\mathbf{Y}),(\hat{\mathbf{X}},\hat{\mathbf{Y}})\right) \sim \Gamma} \left[ c\big((\mathbf{X},\mathbf{Y}), (\hat{\mathbf{X}}, \hat{\mathbf{Y}})\big) \right], \quad (7)$$

where $c\big((\mathbf{X},\mathbf{Y}), (\hat{\mathbf{X}}, \hat{\mathbf{Y}})\big) : (\mathcal{X}, \mathcal{Y}) \times (\mathcal{X}, \mathcal{Y}) \to \mathcal{R}_+$ is any measurable *cost function*. $\mathcal{P}\big((\mathbf{X},\mathbf{Y}) \sim P_{\mathbf{X},\mathbf{Y}}, (\hat{\mathbf{X}}, \hat{\mathbf{Y}}) \sim P_{\hat{\mathbf{X}},\hat{\mathbf{Y}}}\big)$ is the set of all joint distributions of $\big((\mathbf{X},\mathbf{Y}), (\hat{\mathbf{X}}, \hat{\mathbf{Y}})\big)$ with marginals $P_{\mathbf{X},\mathbf{Y}}$ and $P_{\hat{\mathbf{X}},\hat{\mathbf{Y}}}$, respectively. Note that $c\big((\mathbf{X},\mathbf{Y}), (\hat{\mathbf{X}}, \hat{\mathbf{Y}})\big) = c_X\big(\mathbf{X}, \hat{\mathbf{X}}\big) + c_Y\big(\mathbf{Y}, \hat{\mathbf{Y}}\big)$.

Next, we denote the set of all joint distributions of $(\mathbf{X}, \mathbf{Y}, \hat{\mathbf{X}}, \hat{\mathbf{Y}}, \mathbf{Z})$ such that $(\mathbf{X}, \mathbf{Y}) \sim P_{\mathbf{X},\mathbf{Y}}$, $(\hat{\mathbf{X}}, \hat{\mathbf{Y}}, \mathbf{Z}) \sim P_{\hat{\mathbf{X}},\hat{\mathbf{Y}},\mathbf{Z}}$, and $\big((\mathbf{X}, \mathbf{Y}) \perp\!\!\!\perp (\hat{\mathbf{X}}, \hat{\mathbf{Y}})|\mathbf{Z}\big)$ as $\mathcal{P}_{\mathbf{X},\mathbf{Y},\hat{\mathbf{X}},\hat{\mathbf{Y}},\mathbf{Z}}$. $\mathcal{P}_{\mathbf{X},\mathbf{Y},\hat{\mathbf{X}},\hat{\mathbf{Y}}}$ and $\mathcal{P}_{\mathbf{X},\mathbf{Y},\mathbf{Z}}$ are the sets of the marginals $(\mathbf{X}, \mathbf{Y}, \hat{\mathbf{X}}, \hat{\mathbf{Y}})$ and $(\mathbf{X}, \mathbf{Y}, \mathbf{Z})$ induced by $\mathcal{P}_{\mathbf{X},\mathbf{Y},\hat{\mathbf{X}},\hat{\mathbf{Y}},\mathbf{Z}}$.

We now introduce two Lemmas to help the proof.

**Lemma 1.** $P(\hat{\mathbf{X}}, \hat{\mathbf{Y}}|\mathbf{Z} = z)$ *are Dirac for all* $z \in \mathcal{Z}$.

*Proof:* First, we have $\hat{\mathbf{X}} = F(G_a(\mathbf{Z_a}), G_y(\mathbf{Z_y}))$ and $\hat{\mathbf{Y}} = D(G_y(\mathbf{Z_y}))$ with $\mathbf{Z} = \{\mathbf{Z_a}, \mathbf{Z_y}\}$. Since the functions $F, G_a, G_y, D$ are all deterministic, then $P(\hat{\mathbf{X}}, \hat{\mathbf{Y}}|\mathbf{Z})$ are Dirac measures. □

**Lemma 2.** $\mathcal{P}\big(P_{\mathbf{X},\mathbf{Y}}, P_{\hat{\mathbf{X}},\hat{\mathbf{Y}}}\big) = \mathcal{P}_{\mathbf{X},\mathbf{Y},\hat{\mathbf{X}},\hat{\mathbf{Y}}}$ *when* $P(\hat{\mathbf{X}}, \hat{\mathbf{Y}}|\mathbf{Z} = z)$ *are Dirac for all* $z \in \mathcal{Z}$.

*Proof:* When $\hat{\mathbf{X}}, \hat{\mathbf{Y}}$ are deterministic functions of $\mathbf{Z}$, for any $A$ in the sigma-algebra induced by $\hat{\mathbf{X}}, \hat{\mathbf{Y}}$, we have

$$\mathbf{E}[\mathbb{I}_{[\hat{\mathbf{X}}, \hat{\mathbf{Y}} \in A]}|\mathbf{X}, \mathbf{Y}, \mathbf{Z}] = \mathbf{E}[\mathbb{I}_{[\hat{\mathbf{X}}, \hat{\mathbf{Y}} \in A]}|\mathbf{Z}].$$

Therefore, this implies that $(\mathbf{X}, \mathbf{Y}) \perp\!\!\!\perp (\hat{\mathbf{X}}, \hat{\mathbf{Y}})|\mathbf{Z}$ which concludes the proof. A similar argument is made in Lemma 1 of (Tolstikhin et al., 2017).

□

Now, we use the fact that $\mathcal{P}\big(P_{\mathbf{X},\mathbf{Y}}, P_{\hat{\mathbf{X}},\hat{\mathbf{Y}}}\big) = \mathcal{P}_{\mathbf{X},\mathbf{Y},\hat{\mathbf{X}},\hat{\mathbf{Y}}}$ (Lemma 1 + Lemma 2), $c\big((\mathbf{X},\mathbf{Y}),(\hat{\mathbf{X}},\hat{\mathbf{Y}})\big) = c_X\big(\mathbf{X},\hat{\mathbf{X}}\big) + c_Y\big(\mathbf{Y},\hat{\mathbf{Y}}\big)$, $\hat{\mathbf{X}} = F\big(G_a(\mathbf{Z_a}), G_y(\mathbf{Z_y})\big)$, and $\hat{\mathbf{Y}} = D\big(G_y(\mathbf{Z_y})\big)$, Eq. equation 7 becomes

$$
\begin{aligned}
&\inf_{P \in \mathcal{P}_{\mathbf{X},\mathbf{Y},\hat{\mathbf{X}},\hat{\mathbf{Y}}}} \mathbf{E}_{\mathbf{X},\mathbf{Y},\hat{\mathbf{X}},\hat{\mathbf{Y}} \sim P}\Big[c_X\big(\mathbf{X},\hat{\mathbf{X}}\big) + c_Y\big(\mathbf{Y},\hat{\mathbf{Y}}\big)\Big] \\
&= \inf_{P \in \mathcal{P}_{\mathbf{X},\mathbf{Y},\hat{\mathbf{X}},\hat{\mathbf{Y}},\mathbf{Z}}} \mathbf{E}_{\mathbf{X},\mathbf{Y},\hat{\mathbf{X}},\hat{\mathbf{Y}},\mathbf{Z} \sim P}\Big[c_X\big(\mathbf{X},\hat{\mathbf{X}}\big) + c_Y\big(\mathbf{Y},\hat{\mathbf{Y}}\big)\Big] \\
&= \inf_{P \in \mathcal{P}_{\mathbf{X},\mathbf{Y},\hat{\mathbf{X}},\hat{\mathbf{Y}},\mathbf{Z}}} \mathbf{E}_{P_{\mathbf{Z}}} \mathbf{E}_{P(\mathbf{X},\mathbf{Y}|\mathbf{Z})} \mathbf{E}_{P(\hat{\mathbf{X}},\hat{\mathbf{Y}}|\mathbf{Z})}\Big[c_X\big(\mathbf{X},\hat{\mathbf{X}}\big) + c_Y\big(\mathbf{Y},\hat{\mathbf{Y}}\big)\Big] \\
&= \inf_{P \in \mathcal{P}_{\mathbf{X},\mathbf{Y},\hat{\mathbf{X}},\hat{\mathbf{Y}},\mathbf{Z}}} \mathbf{E}_{P_{\mathbf{Z}}} \mathbf{E}_{P(\mathbf{X},\mathbf{Y}|\mathbf{Z})}\Big[c_X\big(\mathbf{X},F\big(G_a(\mathbf{Z_a}),G_y(\mathbf{Z_y})\big)\big) + c_Y\big(\mathbf{Y},D\big(G_y(\mathbf{Z_y})\big)\big)\Big] \\
&= \inf_{P \in \mathcal{P}_{\mathbf{X},\mathbf{Y},\mathbf{Z}}} \mathbf{E}_{P_{\mathbf{Z}}} \mathbf{E}_{P(\mathbf{X},\mathbf{Y}|\mathbf{Z})}\Big[c_X\big(\mathbf{X},F\big(G_a(\mathbf{Z_a}),G_y(\mathbf{Z_y})\big)\big) + c_Y\big(\mathbf{Y},D\big(G_y(\mathbf{Z_y})\big)\big)\Big] \\
&= \inf_{P \in \mathcal{P}_{\mathbf{X},\mathbf{Y},\mathbf{Z}}} \mathbf{E}_{\mathbf{X},\mathbf{Y},\mathbf{Z} \sim P}\Big[c_X\big(\mathbf{X},F\big(G_a(\mathbf{Z_a}),G_y(\mathbf{Z_y})\big)\big) + c_Y\big(\mathbf{Y},D\big(G_y(\mathbf{Z_y})\big)\big)\Big].
\end{aligned}
\tag{8}
$$

Note that in Eq. equation 8, $\mathcal{P}_{\mathbf{X},\mathbf{Y},\mathbf{Z}} = \mathcal{P}\big((\mathbf{X},\mathbf{Y}) \sim P_{\mathbf{X},\mathbf{Y}}, \mathbf{Z} \sim P_{\mathbf{Z}}\big)$ and with a proposed $Q(\mathbf{Z}|\mathbf{X})$, we can rewrite Eq. equation 8 as

$$
\begin{aligned}
&\inf_{P \in \mathcal{P}_{\mathbf{X},\mathbf{Y},\mathbf{Z}}} \mathbf{E}_{P_{\mathbf{X},\mathbf{Y}}} \mathbf{E}_{P_{\mathbf{Z}}}\Big[c_X\big(\mathbf{X},F\big(G_a(\mathbf{Z_a}),G_y(\mathbf{Z_y})\big)\big) + c_Y\big(\mathbf{Y},D\big(G_y(\mathbf{Z_y})\big)\big)\Big] \\
&= \inf_{Q_{\mathbf{Z}} = P_{\mathbf{Z}}} \mathbf{E}_{P_{\mathbf{X},\mathbf{Y}}} \mathbf{E}_{Q(\mathbf{Z}|\mathbf{X})}\Big[c_X\big(\mathbf{X},F\big(G_a(\mathbf{Z_a}),G_y(\mathbf{Z_y})\big)\big) + c_Y\big(\mathbf{Y},D\big(G_y(\mathbf{Z_y})\big)\big)\Big].
\end{aligned}
\tag{9}
$$

∎

## A.2   FROM UNIMODAL TO MULTIMODAL

The proof is similar to Proposition 2, and we present a sketch to it. We can first show $P(\hat{\mathbf{X}}_{1:M}, \hat{\mathbf{Y}}|\mathbf{Z} = z)$ are Dirac for all $z \in \mathcal{Z}$. Then we use the fact that $c\big((\mathbf{X}_{1:M},\mathbf{Y}),(\hat{\mathbf{X}}_{1:M},\hat{\mathbf{Y}})\big) = \sum_{i=1}^{M} c_{X_i}\big(\mathbf{X}_i,\hat{\mathbf{X}}_i\big) + c_Y\big(\mathbf{Y},\hat{\mathbf{Y}}\big)$. Finally, we follow the tower rule of expectation and the conditional independence property similar to the proof in Proposition 2 and this concludes the proof.

∎

## B   FULL BASELINE MODELS & RESULTS

For a detailed description of the baselines, we point the reader to MFN (Zadeh et al., 2018a), MARN (Zadeh et al., 2018b), TFN (Zadeh et al., 2017), BC-LSTM (Poria et al., 2017), MV-LSTM (Rajagopalan et al., 2016), EF-LSTM (Hochreiter & Schmidhuber, 1997; Graves et al., 2013; Schuster & Paliwal, 1997), DF (Nojavanasghari et al., 2016), MV-HCRF (Song et al., 2012; 2013), EF-HCRF (Quattoni et al., 2007; Morency et al., 2007), THMM (Morency et al., 2011), SVM-MD (Zadeh et al., 2016) and RF (Breiman, 2001).

We use the following extra notations for full descriptions of the baseline models described in Section 3.2, paragraph 3:

Variants of EF-LSTM: **EF-LSTM** (Early Fusion LSTM) uses a single LSTM (Hochreiter & Schmidhuber, 1997) on concatenated multimodal inputs. We also implement the **EF-SLSTM** (stacked) (Graves et al., 2013), **EF-BLSTM** (bidirectional) (Schuster & Paliwal, 1997) and **EF-SBLSTM** (stacked bidirectional) versions.

Variants of EF-HCRF: **EF-HCRF**: (Hidden Conditional Random Field) (Quattoni et al., 2007) uses a HCRF to learn a set of latent variables conditioned on the concatenated input at each time step.

**EF-LDHCRF** (Latent Discriminative HCRFs) (Morency et al., 2007) are a class of models that learn hidden states in a CRF using a latent code between observed concatenated input and hidden output. **EF-HSSHCRF**: (Hierarchical Sequence Summarization HCRF) (Song et al., 2013) is a layered model that uses HCRFs with latent variables to learn hidden spatio-temporal dynamics.

Variants of MV-HCRF: **MV-HCRF**: Multi-view HCRF (Song et al., 2012) is an extension of the HCRF for Multi-view data, explicitly capturing view-shared and view specific sub-structures. **MV-LDHCRF**: (Morency et al., 2007) is a variation of the MV-HCRF model that uses LDHCRF instead of HCRF. **MV-HSSHCRF**: (Song et al., 2013) further extends **EF-HSSHCRF** by performing Multi-view hierarchical sequence summary representation.

In the following, we provide the full results for all baselines models described in Section 3.2, paragraph 3. Table 3 contains results for multimodal speaker traits recognition on the POM dataset. Table 4 contains results for the multimodal sentiment analysis on the CMU-MOSI, ICT-MMMO, YouTube, and MOUD datasets. Table 5 contains results for multimodal emotion recognition on the IEMOCAP dataset. MFM consistently achieves state-of-the-art or competitive results for all six multimodal datasets. We believe that by our MFM design, the multimodal discriminative factor $\mathbf{F_y}$ has successfully learned more meaningful representations by distilling discriminative features. This highlights the benefit of learning factorized multimodal representations towards discriminative tasks.

## C  MULTIMODAL FEATURES

For each of the multimodal time series datasets as mentioned in Section 3.2, paragraph 3, we extracted the following multimodal features: **Language:** We use pre-trained word embeddings (glove.840B.300d) (Pennington et al., 2014) to convert the video transcripts into a sequence of 300 dimensional word vectors. **Visual:** We use Facet (iMotions, 2017) to extract a set of features including per-frame basic and advanced emotions and facial action units as indicators of facial muscle movement (Ekman et al., 1980; Ekman, 1992). **Acoustic:** We use COVAREP (Degottex et al., 2014) to extract low level acoustic features including 12 Mel-frequency cepstral coefficients (MFCCs), pitch tracking and voiced/unvoiced segmenting features, glottal source parameters, peak slope parameters and maxima dispersion quotients. To reach the same time alignment between different modalities we choose the granularity of the input to be at the level of words. The words are aligned with audio using P2FA (Yuan & Liberman, 2008) to get their exact utterance times. We use expected feature values across the entire word for visual and acoustic features since they are extracted at a higher frequencies.

We make a note that the features for some of these datasets are constantly being updated. The authors of Zadeh et al. (2018a) notified us of a discrepancy in the sampling rate for acoustic feature extraction in the ICT-MMMO, YouTube and MOUD datasets which led to inaccurate word-level alignment between the three modalities. They publicly released the updated multimodal features. We performed all experiments on the latest versions of these datasets which can be accessed from `https://github.com/A2Zadeh/CMU-MultimodalSDK`. All baseline models were retrained with extensive hyperparameter search for fair comparison.

## D  INFORMATION AND GRADIENT-BASED INTERPRETATION

**Information-Based Interpretation:** We choose the normalized Hilbert-Schmidt Independence Criterion (Gretton et al., 2005; Wu et al., 2018) as the approximation (see Sugiyama & Yamada (2012); Wu et al. (2018)) of our MI measure:

$$\text{MI}(\mathbf{F}_{\cdot}, \hat{\mathbf{X}}_i) = \text{HSIC}_{norm}(\mathbf{F}_{\cdot}, \hat{\mathbf{X}}_i) = \frac{\text{tr}(\mathbf{K_{F_{\cdot}}HK_{\hat{X}_i}H})}{\|\mathbf{HK_{F_{\cdot}}H}\|_F\|\mathbf{HK_{\hat{X}_i}H}\|_F}, \tag{10}$$

where $\cdot$ represents $y$ or $a_i$, $n$ is the number of $\{\mathbf{F}_{\cdot}, \hat{\mathbf{X}}_i\}$ pairs, $\mathbf{H} = \mathbf{I} - \frac{1}{n}\mathbf{1}\mathbf{1}^\top$, $\mathbf{K_{F_{\cdot}}} \in \mathbb{R}^{n \times n}$ is the Gram matrix of $\mathbf{F}_{\cdot}$ with $\mathbf{K}_{\mathbf{F}_{\cdot}ij} = k_1(\mathbf{F}_{\cdot i}, \mathbf{F}_{\cdot j})$, $\mathbf{K}_{\hat{\mathbf{X}}_i} \in \mathbb{R}^{n \times n}$ is the Gram matrix of $\hat{\mathbf{X}}_i$ with $\mathbf{K}_{\hat{\mathbf{X}}_ijk} = k_2(\hat{\mathbf{X}}_{ij}, \hat{\mathbf{X}}_{ik})$. $k_1(\cdot, \cdot)$ and $k_2(\cdot, \cdot)$ are predefined kernel functions.

The most common choice for the kernel is the RBF kernel. However, if we consider time series data with various time steps, we need to either perform data augmentation or choose another kernel choice. For example, we can adopt the Global Alignment Kernel (Cuturi et al., 2007) which considers the

Table 3: Results for personality trait recognition on the POM dataset. The best results are highlighted in bold and $\Delta_{SOTA}$ shows the change in performance over previous state of the art. Improvements are highlighted in green. MFM achieves state-of-the-art or competitive performance on all datasets and metrics.

| Dataset | POM Speaker Personality Traits | | | | | | | | | | | | | | | |
|---|---|---|---|---|---|---|---|---|---|---|---|---|---|---|---|---|
| Task | Con | Pas | Voi | Dom | Cre | Viv | Exp | Ent | Res | Tru | Rel | Out | Tho | Ner | Per | Hum |
| Metric | | | | | | | | | $r$ | | | | | | | |
| Majority | -0.041 | -0.029 | -0.104 | -0.031 | -0.122 | -0.044 | -0.065 | -0.105 | 0.006 | -0.077 | -0.024 | -0.085 | -0.130 | 0.097 | -0.127 | -0.069 |
| SVM | 0.063 | 0.086 | -0.004 | 0.141 | 0.113 | 0.076 | 0.134 | 0.141 | 0.166 | 0.168 | 0.104 | 0.066 | 0.134 | 0.068 | 0.064 | 0.147 |
| DF | 0.240 | 0.273 | 0.017 | 0.139 | 0.112 | 0.173 | 0.118 | 0.217 | 0.148 | 0.143 | 0.019 | 0.093 | 0.041 | 0.136 | 0.168 | 0.259 |
| EF-LSTM | 0.200 | 0.302 | 0.031 | 0.079 | 0.170 | 0.244 | 0.265 | 0.240 | 0.142 | 0.062 | 0.083 | 0.152 | 0.260 | 0.105 | 0.217 | 0.227 |
| EF-SLSTM | 0.221 | 0.327 | 0.042 | 0.151 | 0.177 | 0.239 | 0.268 | 0.248 | 0.204 | 0.069 | 0.092 | 0.215 | 0.252 | 0.159 | 0.218 | 0.196 |
| EF-BLSTM | 0.162 | 0.289 | -0.034 | 0.135 | 0.191 | 0.279 | 0.274 | 0.231 | 0.184 | 0.154 | 0.093 | 0.147 | 0.245 | 0.166 | 0.243 | 0.272 |
| EF-SBLSTM | 0.174 | 0.310 | 0.021 | 0.088 | 0.170 | 0.224 | 0.261 | 0.241 | 0.155 | 0.163 | 0.097 | 0.120 | 0.215 | 0.121 | 0.216 | 0.171 |
| MV-LSTM | 0.358 | 0.416 | 0.131 | 0.146 | 0.280 | 0.347 | 0.323 | 0.326 | 0.295 | 0.237 | 0.119 | 0.238 | 0.284 | 0.258 | 0.239 | 0.317 |
| BC-LSTM | 0.359 | 0.425 | 0.081 | 0.234 | 0.358 | 0.417 | 0.450 | 0.361 | 0.293 | 0.109 | 0.075 | 0.078 | 0.363 | 0.184 | 0.344 | 0.319 |
| TFN | 0.089 | 0.201 | 0.030 | 0.020 | 0.124 | 0.204 | 0.171 | 0.223 | -0.051 | -0.064 | 0.114 | 0.060 | 0.048 | -0.002 | 0.106 | 0.213 |
| MARN | 0.340 | 0.410 | 0.166 | 0.235 | 0.340 | 0.374 | 0.406 | 0.378 | 0.282 | 0.147 | 0.215 | 0.204 | 0.348 | 0.235 | 0.303 | 0.287 |
| MFN | 0.395 | 0.428 | 0.193 | 0.313 | 0.367 | 0.431 | 0.452 | 0.395 | 0.333 | 0.296 | 0.255 | 0.259 | 0.381 | 0.318 | 0.377 | 0.386 |
| MFM | **0.431** | **0.450** | **0.197** | **0.411** | **0.380** | **0.448** | **0.467** | **0.452** | **0.368** | 0.212 | **0.309** | **0.333** | **0.404** | **0.333** | 0.334 | **0.408** |
| $\Delta_{SOTA}$ | ↑ 0.036 | ↑ 0.022 | ↑ 0.004 | ↑ 0.097 | ↑ 0.013 | ↑ 0.017 | ↑ 0.015 | ↑ 0.057 | ↑ 0.035 | – | ↑ 0.054 | ↑ 0.074 | ↑ 0.023 | ↑ 0.015 | – | ↑ 0.022 |

Table 4: Sentiment prediction results on CMU-MOSI, ICT-MMMO, YouTube and MOUD. The best results are highlighted in bold and $\Delta_{SOTA}$ shows the change in performance over previous state of the art (SOTA). Improvements are highlighted in green. MFM achieves state-of-the-art or competitive performance on all datasets and metrics.

| Dataset | CMU-MOSI | | | | | ICT-MMMO | | YouTube | | MOUD | |
|---|---|---|---|---|---|---|---|---|---|---|---|
| Task | Sentiment | | | | | Sentiment | | Sentiment | | Sentiment | |
| Metric | Acc_7 | Acc_2 | F1 | MAE | $r$ | Acc_2 | F1 | Acc_3 | F1 | Acc_2 | F1 |
| Majority | 17.5 | 50.2 | 50.1 | 1.864 | 0.057 | 40.0 | 22.9 | 42.4 | 25.2 | 60.4 | 45.5 |
| RF | 21.3 | 56.4 | 56.3 | - | - | 70.0 | 69.8 | 33.3 | 32.3 | 64.2 | 63.3 |
| SVM-MD | 26.5 | 71.6 | 72.3 | 1.100 | 0.559 | 68.8 | 68.7 | 42.4 | 37.9 | 59.4 | 45.5 |
| THMM | 17.8 | 53.8 | 53.0 | - | - | 50.7 | 45.4 | 42.4 | 27.9 | 61.3 | 57.0 |
| SAL-CNN | - | 73.0 | - | - | - | - | - | - | - | - | - |
| C-MKL | 30.2 | 72.3 | 72.0 | - | - | - | - | - | - | - | - |
| EF-HCRF | 24.6 | 65.3 | 65.4 | - | - | 50.0 | 50.3 | 44.1 | 43.8 | 54.7 | 54.7 |
| EF-LDHCRF | 24.6 | 64.0 | 64.0 | - | - | 73.8 | 73.1 | 45.8 | 45.0 | 52.8 | 49.3 |
| MV-HCRF | 22.6 | 44.8 | 27.7 | - | - | 36.3 | 19.3 | 27.1 | 19.7 | 60.4 | 45.5 |
| MV-LDHCRF | 24.6 | 64.0 | 64.0 | - | - | 68.8 | 67.1 | 44.1 | 44.0 | 53.8 | 46.9 |
| CMV-HCRF | 22.3 | 44.8 | 27.7 | - | - | 36.3 | 19.3 | 30.5 | 14.3 | 60.4 | 45.5 |
| CMV-LDHCRF | 24.6 | 63.6 | 63.6 | - | - | 51.3 | 51.4 | 42.4 | 42.0 | 53.8 | 47.8 |
| EF-HSSHCRF | 24.6 | 63.3 | 63.4 | - | - | 50.0 | 51.3 | 37.3 | 35.6 | 52.8 | 49.3 |
| MV-HSSHCRF | 24.6 | 65.6 | 65.7 | - | - | 62.5 | 63.1 | 44.1 | 44.0 | 47.2 | 46.4 |
| DF | 26.8 | 72.3 | 72.1 | 1.143 | 0.518 | 65.0 | 58.7 | 45.8 | 32.0 | 67.0 | 67.1 |
| EF-LSTM | 32.4 | 74.3 | 74.3 | 1.023 | 0.622 | 66.3 | 65.0 | 44.1 | 43.6 | 67.0 | 64.3 |
| EF-SLSTM | 29.3 | 72.7 | 72.8 | 1.081 | 0.600 | 72.5 | 70.9 | 40.7 | 41.2 | 56.6 | 51.4 |
| EF-BLSTM | 28.9 | 72.0 | 72.0 | 1.080 | 0.577 | 63.8 | 49.6 | 42.4 | 38.1 | 58.5 | 58.9 |
| EF-SBLSTM | 26.8 | 73.3 | 73.2 | 1.037 | 0.619 | 62.5 | 49.0 | 37.3 | 33.2 | 63.2 | 63.3 |
| MV-LSTM | 33.2 | 73.9 | 74.0 | 1.019 | 0.601 | 72.5 | 72.3 | 45.8 | 43.3 | 57.6 | 48.2 |
| BC-LSTM | 28.7 | 73.9 | 73.9 | 1.079 | 0.581 | 70.0 | 70.1 | 45.0 | 45.1 | 72.6 | 72.9 |
| TFN | 28.7 | 74.6 | 74.5 | 1.040 | 0.587 | 72.5 | 72.6 | 45.0 | 41.0 | 63.2 | 61.7 |
| MARN | 34.7 | 77.1 | 77.0 | 0.968 | 0.625 | 71.3 | 70.2 | 48.3 | 44.9 | 81.1 | 81.2 |
| MFN | 34.1 | 77.4 | 77.3 | 0.965 | 0.632 | 73.8 | 73.1 | 51.7 | 51.6 | 81.1 | 80.4 |
| MFM | **36.2** | **78.1** | **78.1** | **0.951** | **0.662** | **81.3** | **79.2** | **53.3** | **52.4** | **82.1** | **81.7** |
| $\Delta_{SOTA}$ | ↑ 1.5 | ↑ 0.7 | ↑ 0.8 | ↓ 0.014 | ↑ 0.030 | ↑ 7.5 | ↑ 6.1 | ↑ 1.6 | ↑ 0.8 | ↑ 1.0 | ↑ 0.5 |

Table 5: Emotion recognition results on IEMOCAP test set. The best results are highlighted in bold and $\Delta_{SOTA}$ shows the change in performance over previous SOTA. Improvements are highlighted in green. MFM achieves state-of-the-art or competitive performance on all datasets and metrics.

| Dataset | **IEMOCAP Emotions** | | | | | | | | | | | |
|---|---|---|---|---|---|---|---|---|---|---|---|---|
| Task | Happy | | Sad | | Angry | | Frustrated | | Excited | | Neutral | |
| Metric | Acc_2 | F1 | Acc_2 | F1 | Acc_2 | F1 | Acc_2 | F1 | Acc_2 | F1 | Acc_2 | F1 |
| Majority | 85.6 | 79.0 | 79.4 | 70.3 | 75.8 | 65.4 | 79.5 | 70.4 | 89.6 | 84.7 | 59.1 | 44.0 |
| SVM | 86.1 | 81.5 | 81.1 | 78.8 | 82.5 | 82.4 | 77.3 | 71.1 | 86.4 | 86.0 | 65.2 | 64.9 |
| RF | 85.5 | 80.7 | 80.1 | 76.5 | 81.9 | 82.0 | 78.6 | 75.3 | 88.9 | 85.1 | 63.2 | 57.3 |
| THMM | 85.6 | 79.2 | 79.5 | 79.8 | 79.3 | 73.0 | 71.6 | 69.6 | 86.0 | 84.6 | 58.6 | 46.4 |
| EF-HCRF | 85.7 | 79.2 | 79.4 | 70.3 | 75.8 | 65.4 | 79.5 | 70.4 | 89.6 | 84.7 | 59.1 | 44.0 |
| EF-LDHCRF | 85.8 | 79.5 | 79.4 | 70.3 | 75.8 | 65.4 | 79.5 | 70.4 | 89.6 | 84.7 | 59.1 | 44.0 |
| MV-HCRF | 15.0 | 4.9 | 79.4 | 70.3 | 24.2 | 9.4 | 79.5 | 70.4 | 89.6 | 84.7 | 59.1 | 44.0 |
| MV-LDHCRF | 85.7 | 79.2 | 79.4 | 70.3 | 75.8 | 65.4 | 79.5 | 70.4 | 89.6 | 84.7 | 59.1 | 44.0 |
| CMV-HCRF | 14.4 | 3.6 | 79.4 | 70.3 | 24.2 | 9.4 | 79.5 | 70.4 | 89.6 | 84.7 | 59.1 | 44.0 |
| CMV-LDHCRF | 85.8 | 79.5 | 79.4 | 70.3 | 75.8 | 65.4 | 79.5 | 70.4 | 89.6 | 84.7 | 59.1 | 44.0 |
| EF-HSSHCRF | 85.8 | 79.5 | 79.4 | 70.3 | 75.8 | 65.4 | 79.5 | 70.4 | 89.6 | 84.7 | 59.1 | 44.0 |
| MV-HSSHCRF | 85.8 | 79.5 | 79.4 | 70.3 | 75.8 | 65.4 | 79.5 | 70.4 | 89.6 | 84.7 | 59.1 | 44.0 |
| DF | 86.0 | 81.0 | 81.8 | 81.2 | 75.8 | 65.4 | 78.4 | 76.8 | 89.6 | 84.7 | 59.1 | 44.0 |
| EF-LSTM | 85.2 | 83.3 | 82.1 | 81.1 | 84.5 | 84.3 | 79.5 | 70.4 | 89.6 | 84.7 | 68.2 | 67.1 |
| EF-SLSTM | 85.6 | 79.0 | 80.7 | 80.2 | 82.8 | 82.2 | 77.5 | 69.7 | 89.3 | 86.2 | 68.8 | **68.5** |
| EF-BLSTM | 85.0 | 83.7 | 81.8 | 81.6 | 84.2 | 83.3 | 79.5 | 70.4 | 89.6 | 84.7 | 67.1 | 66.6 |
| EF-SBLSTM | 86.0 | 84.2 | 80.2 | 80.5 | 85.2 | 84.5 | 79.5 | 70.4 | 89.6 | 84.7 | 67.8 | 67.1 |
| MV-LSTM | 85.9 | 81.3 | 80.4 | 74.0 | 85.1 | 84.3 | 79.5 | 73.8 | 88.9 | 85.8 | 67.0 | 66.7 |
| BC-LSTM | 84.9 | 81.7 | 83.2 | 81.7 | 83.5 | 84.2 | 80.0 | 76.1 | 86.9 | 85.4 | 67.5 | 64.1 |
| TFN | 84.8 | 83.6 | 83.4 | 82.8 | 83.4 | 84.2 | 74.1 | 74.3 | 75.6 | 78.0 | 67.5 | 65.4 |
| MARN | 86.7 | 83.6 | 82.0 | 81.2 | 84.6 | 84.2 | 79.5 | 76.6 | 89.6 | **87.1** | 66.8 | 65.9 |
| MFN | 90.1 | 85.3 | 85.8 | 79.2 | 87.0 | 86.0 | 80.3 | **76.9** | 89.8 | 86.3 | 71.8 | 61.7 |
| MFM | **90.2** | **85.8** | **88.4** | **86.1** | **87.5** | **86.7** | 80.4 | 74.5 | **90.0** | 87.1 | **72.1** | 68.1 |
| $\Delta_{SOTA}$ | ↑ 0.1 | ↑ 0.5 | ↑ 2.6 | ↑ 3.3 | ↑ 0.5 | ↑ 0.7 | ↑ 0.1 | – | ↑ 0.2 | – | ↑ 0.3 | – |

alignment between two varying-length time series when computing the kernel score between them. To simplify our analysis, we choose to augment data before we calculate the kernel score with the RBF kernel. More specifically, we perform averaging over time series data:

$$\mathbf{X}_{aug} = \frac{1}{n} \sum_{t=1}^{T} X^t \text{ with } \mathbf{X} = [X^1, X^2, \cdots, X^T]. \tag{11}$$

The bandwidth of the RBF kernel is set as 1.0 throughout the experiments.

Table 6: Information-Based interpretation results showing ratios $r_i = \frac{\text{MI}(\mathbf{F_y}, \hat{\mathbf{X}}_i)}{\text{MI}(\mathbf{F_{a}}_i, \hat{\mathbf{X}}_i)}$, $i \in \{(\ell)anguage, (v)isual, (a)coustic\}$ for the POM dataset for personality traits prediction.

| Ratio | $r_\ell$ (language) | $r_v$ (visual) | $r_a$ (acoustic) |
|---|---|---|---|
| POM | 1.090 | 0.996 | 0.898 |

Here, we provide an additional interpretation result for the POM dataset in Table 6. We observe that the language modality is also the most informative while the visual and acoustic modalities are almost equally informative. This result is in agreement with behavioral studies which have observed that non-verbal behaviors are particularly informative of personality traits (Guimond & Massrieh, 2012; Levine et al., 2009; Mohammadi et al., 2010). For example, the same sentence "this movie was great" can convey significantly different messages on speaker confidence depending on whether it was said in a loud and exciting voice, with eye contact, or powerful gesticulation.

**Gradient-Based Interpretation:** MFM reconstructs $x_i$ as follows:

$$\hat{x}_i = F_i(f_{ai}, f_y), f_{ai} = G_{ai}(z_{ai}), f_y = G_y(z_y), z_{ai} \sim Q(\mathbf{Z_{a}}_i|\mathbf{X}_i = x_i), z_y \sim Q(\mathbf{Z_y}|\mathbf{X}_{1:M} = x_{1:M}). \tag{12}$$

Equation equation 12 also explains how we obtain $f_y \sim P(\mathbf{F_y}|\mathbf{X}_{1:M} = x_{1:M})$. The gradient flow through time is defined as:

$$\nabla_{f_y}(\hat{x}_i) := [\|\nabla_{f_y}\hat{x}_i^1\|_F^2, \|\nabla_{f_y}\hat{x}_i^2\|_F^2, \cdots, \|\nabla_{f_y}\hat{x}_i^T\|_F^2]. \tag{13}$$

## E    ENCODER AND DECODER DESIGN FOR MULTIMODAL SYNTHETIC IMAGE DATASET

For experiments on the multimodal synthetic image dataset, we use convolutional+fully-connected layers for the encoder and deconvolutional+fully-connected layers for the decoder (Zeiler et al., 2010). Different convolutional layers are each applied on the input SVHN and MNIST images to learn modality-specific generative factors. Next, we concatenate the features from two more convolutional layers on SVHN and MNIST to learn the multimodal-discriminative factor. The multimodal discriminative factor is passed through fully-connected layers to predict the label. For generation, we concatenate the multimodal discriminative factors and the modality-specific generative factor together and use a deconvolutional layer to generate digits.

## F    ENCODER AND DECODER DESIGN FOR MULTIMODAL TIME SERIES DATASETS

Figure 5 illustrates how MFM operates on multimodal time series data. The encoder $Q(\mathbf{Z_y}|\mathbf{X}_{1:M})$ can be parametrized by any model that performs multimodal fusion (Nojavanasghari et al., 2016; Zadeh et al., 2018a). We choose the Memory Fusion Network (MFN) (Zadeh et al., 2018a) as our encoder $Q(\mathbf{Z_y}|\mathbf{X}_{1:M})$. We use encoder LSTM networks and decoder LSTM networks (Cho et al., 2014) to parametrize functions $Q(\mathbf{Z_{a}}_{1:M}|\mathbf{X}_{1:M})$ and $F_{1:M}$ respectively, and FCNNs to parametrize functions $G_y$, $G_{a\{1:M\}}$ and $D$.

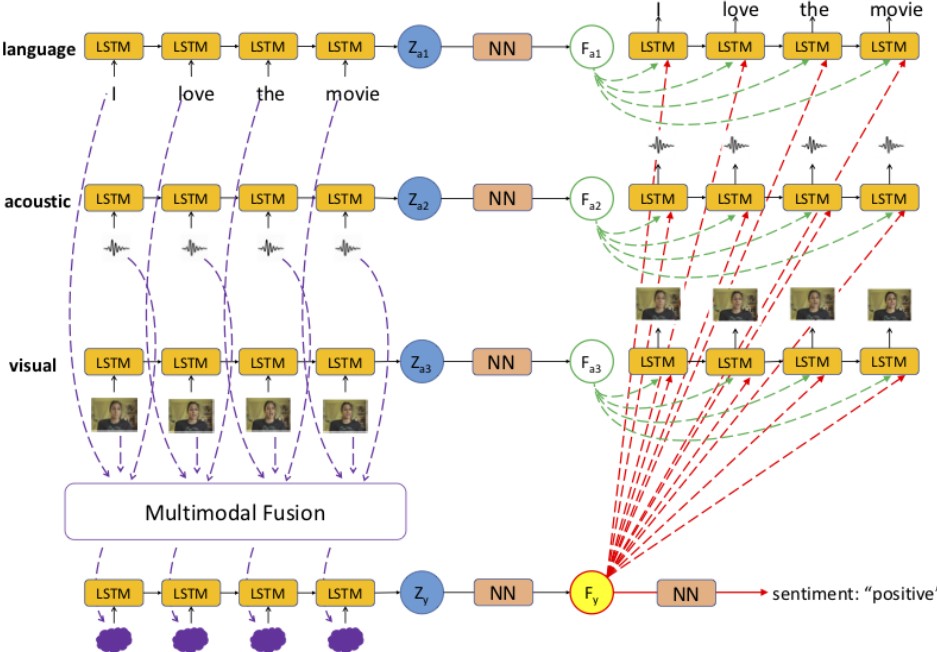

Figure 5: Recurrent neural architecture for MFM. The encoder $Q(\mathbf{Z_y}|\mathbf{X}_{1:M})$ can be parametrized by any model that performs multimodal fusion (Nojavanasghari et al., 2016; Zadeh et al., 2018a). We use encoder LSTM networks and decoder LSTM networks (Cho et al., 2014) to parametrize functions $Q(\mathbf{Z_{a}}_{1:M}|\mathbf{X}_{1:M})$ and $F_{1:M}$ respectively, and FCNNs to parametrize functions $G_y$, $G_{a\{1:M\}}$ and $D$.

## G    SURROGATE INFERENCE GRAPHICAL MODEL

We illustrate the surrogate inference for addressing the missing modalities issue in Figure 6. The surrogate inference model infers the latent codes given the present modalities. These inferred latent codes can then be used for reconstructing the missing modalities or label prediction in the presence of missing modalities.

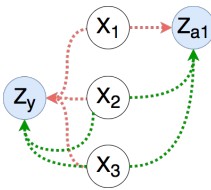

Figure 6: The surrogate inference graphical model to deal with missing modalities in MFM. Red lines denote original inference in MFM and green lines denote surrogate inference to infer latent codes given present modalities.

## H    COMPARISON WITH HSU & GLASS (2018)

A similar approach for factorizing the latent factors was recently proposed by Hsu & Glass (2018) in work that was performed independently and in parallel. In comparison with MFM, there are several major differences that can be categorized into the (1) prior matching discrepancy, (2) inference network, (3) discriminative objective, (4) multimodal fusion, (5) scale of experiments.

1. MFM uses $\mathcal{MMD}(Q_{\mathbf{Z}}, P_{\mathbf{Z}})$ (see Equation 4) as the prior matching discrepancy while Hsu & Glass (2018) use $\mathcal{KL}(Q_{\mathbf{Z}|\mathbf{X}}, P_{\mathbf{Z}})$ (see Section 2.2.1 in Hsu & Glass (2018)).

2. MFM considers multimodal and unimodal inference in a single network (see Figure 1(b)), while Hsu & Glass (2018) considers separate networks (see Figure 1 in Hsu & Glass (2018)). They further propose to match the coherence between these two networks using an additional loss term (see Equation 7 in Hsu & Glass (2018)).

3. MFM learns to predict the labels using a generative framework (see Figure 1(a)), while Hsu & Glass (2018) use an additional hinge loss to separate the latent factors from different labels (see Equation 9 in Hsu & Glass (2018)).

4. MFM is a flexible framework that can be combined with any multimodal fusion encoder (see Section 2.4), while Hsu & Glass (2018) considers a fixed multimodal encoder (similar to early fusion) (see Section 4.1 in Hsu & Glass (2018)).

5. We evaluate the performance of MFM over a much larger scale of datasets. We perform experiments on six multimodal time-series datasets that take on the form of videos with the language, visual, and acoustic modalities. These datasets span three core research areas of multimodal personality traits recognition, multimodal sentiment analysis, and multimodal emotion recognition. On the other hand, Hsu & Glass (2018) evaluates their model on a spoken digit dataset which randomly combines a digit image with a spoken digit (see Section 4 in Hsu & Glass (2018)). MFM further considers experiments to evaluate reconstruction and prediction in the presence of missing modalities (see Section 3.2) which Hsu & Glass (2018) do not. Lastly, we compares to over 20 baseline models in our experiments (see Section 3.2) and explore the choice of various multimodal encoders in MFM. Hsu & Glass (2018) only compares to the JMVAE baseline model (Suzuki et al., 2016a) which resembles the $\mathbf{M_D}$ model in our ablation study (see Section 4.4 in Hsu & Glass (2018)).

In Table 7, we provide a comparison on the CMU-MOSI, ICT-MMMO, YouTube and MOUD datasets to test the disentanglement and prediction performance for the model described in Hsu & Glass (2018). These experimental results show that across these datasets and metrics, MFM performs better than the model proposed in Hsu & Glass (2018). We would like to highlight that at the time of submission, the code for (Hsu & Glass, 2018) had not been made public and we reimplemented their model to experiment on our datasets.

## I    COMPARISON WITH $\beta$-VAE

Although $\beta$-VAE (Higgins et al., 2017) was designed to handle unimodal data, we provide an extension to multimodal data. To achieve this, we set the choice of prior matching discrepancy as the KL-divergence $\mathcal{KL}(Q_{\mathbf{Z}|\mathbf{X}}, P_{\mathbf{Z}})$ and set $\beta$ large (i.e. $\beta \in \{10, 50, 100, 200\}$) to encourage disentanglement of latent variables. We train a $\beta$-VAE to model multimodal data using the same

Table 7: Comparison with Hsu & Glass (2018) for sentiment analysis on CMU-MOSI, ICT-MMMO, YouTube, and MOUD. MFM outperforms the baselines across these datasets and metrics.

| Dataset | CMU-MOSI | | | | | ICT-MMMO | | YouTube | | MOUD | |
|---|---|---|---|---|---|---|---|---|---|---|---|
| Task | Sentiment | | | | | Sentiment | | Sentiment | | Sentiment | |
| Metric | Acc_7 | Acc_2 | F1 | MAE | $r$ | Acc_2 | F1 | Acc_3 | F1 | Acc_2 | F1 |
| without factorization (EF) | 32.4 | 74.3 | 74.3 | 1.023 | 0.622 | 72.5 | 70.9 | 44.1 | 43.6 | 67.0 | 64.3 |
| without factorization (MFN) | 34.1 | 77.4 | 77.3 | 0.965 | 0.632 | 73.8 | 73.1 | 51.7 | 51.6 | 81.1 | 80.4 |
| Hsu & Glass (2018) | 33.8 | 75.2 | 75.2 | 1.049 | 0.584 | 77.5 | 75.0 | 51.7 | 48.6 | 66.0 | 62.9 |
| MFM | **36.2** | **78.1** | **78.1** | **0.951** | **0.662** | **81.3** | **79.2** | **53.3** | **52.4** | **82.1** | **81.7** |

Table 8: Comparison with $\beta$-VAE for multimodal sentiment analysis on CMU-MOSI, ICT-MMMO, YouTube, and MOUD. MFM outperforms $\beta$-VAE across these datasets and metrics.

| Dataset | CMU-MOSI | | | | | ICT-MMMO | | YouTube | | MOUD | |
|---|---|---|---|---|---|---|---|---|---|---|---|
| Task | Sentiment | | | | | Sentiment | | Sentiment | | Sentiment | |
| Metric | Acc_7 | Acc_2 | F1 | MAE | $r$ | Acc_2 | F1 | Acc_3 | F1 | Acc_2 | F1 |
| $\beta$-VAE | 29.7 | 71.3 | 71.3 | 1.094 | 0.552 | 65.0 | 59.8 | 46.7 | 31.3 | 60.4 | 54.2 |
| MFM | **36.2** | **78.1** | **78.1** | **0.951** | **0.662** | **81.3** | **79.2** | **53.3** | **52.4** | **82.1** | **81.7** |

factorization as proposed in our model (i.e. modality-specific generative factors $\mathbf{Z_{a}}_{\{1:M\}}$ and a multimodal discriminative factor $\mathbf{Z_y}$). To provide a fair comparison to our discriminative model, we fine tune by training a classifier on top of the multimodal discriminative factor $\mathbf{Z_y}$ to the label $\mathbf{Y}$. We provide experimental results in Table 8 on the CMU-MOSI, ICT-MMMO, YouTube and MOUD datasets. MFM outperforms $\beta$-VAE across these datasets and metrics.

