# OpenReview forum: "Learning Factorized Multimodal Representations"
_ICLR.cc/2019/Conference_

### Official Review · AnonReviewer2 · 2018-10-28
**Nice Work**

**Rating:** 6
**Confidence:** 3

**Review:**

The authors splitted the features of multimodal representations to "common" (multimodal discriminative) and "specific" (modality-specific generative) factors. In this framework, their MFM can capture more detailed features.

Pros:
(*) Learning the feature representations from two perspectives.

(*) Even missing one modality, MFM can still achieve acceptable performance.

(*) Using mutual information and gradient-based method to interpret their method.

Cons:
(*) The work has some similarity to Hsu & Glass (2018), but the comparison between this work is only on CMU-MOSI.

(*) In Table. 3, it shows that language is the most informative feature for prediction. However, in Table. 2, it can be seen that if audio is missing, the result it the worse compared to the other two cases. It seems the interpretation is not convincing to me. Can you give us more explanation about this phenomenon?

Comments:
(*) The details of SVHN-MNIST experiment are missing. Appendix B gave some information about models but specified the targeted datasets.

(*) The appendix is not clear, e.g. In Appendix B, it is said "subsection 3.3" but there is no section 3.3.

---

> ### Author Response · Authors · 2018-11-15
> **thank you for your feedback!**
>
> Thank you for your positive comments and suggestions for improvement. We address your comments and questions below.
>
> [Comparison with  Hsu & Glass (2018)] We have performed additional experiments between our model and Hsu & Glass (2018) on 3 more multimodal datasets. Our proposed MFM model consistently outperforms the model proposed in Hsu & Glass (2018). In terms of sentiment classification accuracy, we obtain:
> - 3.9% improvement on CMU-MOSI
> - 4.9% improvement on ICT-MMMO
> - 3.1% improvement on YouTube
> - 24.4% improvement on MOUD
> For more details, please refer to full results in appendix H. We would like to emphasize that our present work started in February, and was performed independently and in parallel to the arXiv submission by Hsu & Glass (2018) in July.
>
> We also highlight a few key differences: 1) we use an MMD prior matching discrepancy derived from an extension of WAE from marginal to joint distributions over multimodal data and labels, 2) we train a single network architecture to learn factorized representations, 3) we use a generative-discriminative objective function, and 4) we perform experiments over 6 large-scale multimodal datasets across 3 multimodal tasks. For more details, please refer to appendix section H.
>
> [Language and audio] There seems to be some misunderstanding regarding Table 2: we note that having a missing audio modality does not lead to worse prediction performance. Instead, label prediction performance (Y prediction) suffers the most when the language modality is missing.
> Results in terms of binary classification accuracy (see Table 2):
> - Language missing: 62.0%
> - Audio missing: 74.3%
> - Visual missing: 74.6%
> - All present: 78.1%
> Similar results hold for other metrics as well. Discriminative performance is most affected when the language modality is missing, which is consistent with prior work which indicates that language is most informative in multimodal setting (Zadeh et al., 2017). On the other hand, sentiment prediction is more robust to missing acoustic and visual features.
>
> [SVHN-MNIST experiments] We will release the code along with the appropriate dataset preprocessing details to avoid any ambiguity. Some details are provided in subsection 3.1, multimodal image datasets: Specifically, SVHN and MNIST are images with different styles but the same labels (digits 0 ∼ 9). We randomly pair 100,000 SVHN and MNIST images that have the same label, creating a multimodal dataset which we call SVHN+MNIST. 80,000 pairs are used for training and the rest for testing. To show that MFM is able to learn improved multimodal representations, we provided both classification and generation results on SVHN+MNIST in Figure 2. We use convolution layers to learn the latent codes from images and deconvolution layers to generate images from the latent codes. We have updated the paper with more details (appendix section E).
>
> [Appendix] We apologize for the typo. The baselines models referred to are in subsection 3.2, paragraph 3 (prediction on multimodal time series datasets). We have updated the paper.

---

> > ### Comment · AnonReviewer2 · 2018-12-06
> > **Thanks**
> >
> > Thanks for addressing my confusions.

---

### Official Review · AnonReviewer3 · 2018-11-02
**Good work**

**Rating:** 7
**Confidence:** 2

**Review:**

Multimodality learning is an important topic in multimedia and human computer interaction.  How to efficiently leverage the additional information cross multimodality is the key to the task. Authors proposed the Bayesian latent variable model to factorize the multimodality representation into multimodal discriminative factors and modality-specific generating factors, which is interesting. Approximate inference is also proposed to learn this model via a generalised mean-field assumption.

The technical quality of the paper is sound and significant, The problem to solve in this paper is also well motivated and important.  In general, this is a well-written paper,

I have a few minor questions which requires authors for further elaboration.
1. If I understand it correctly, in the current work, the feature Xs are continuous. Does the approach apply to categorical or binary features?

2. In equation(4), MMD is used. How to solve the computation complexity problem since the complexity of MMD is O(n^2)?  It is true that the batch size should be small?  How to select the hyper-parameters of kernels?

---

> ### Author Response · Authors · 2018-11-15
> **thank you for your feedback!**
>
> Thank you for your positive comments and suggestions for improvement. We address your comments and questions below.
>
> [Categorical or binary features] Yes, the encoders and decoders can be flexibly chosen for continuous or categorical features (e.g. for text, we can use discrete word tokens, represented as one-hot vectors, and train an embedding layer for task-specific word embeddings.) To model categorical or binary features, we can also choose the suitable distance metric designed for them, for example, Jaccard distance.
>
> [MMD computational complexity] In this work, we adopt the unbiased MMD estimator which has a computational complexity of O(n^2). The complexity can be reduced if we choose a block, linear, or incomplete MMD estimator [1]. For example, if we use a linear MMD estimator, the computation time required is O(n), and in the batch setting, it is O(b) where b is the batch size.
>
> [Batchsize] We found that a batchsize of 32, 64 or 128 works well in our experiments.
>
> [Hyperparameters] As suggested by WAE, we first set our RBF kernel bandwidth to be sqrt(d), where d is the dimension of latent variables in WAE. Using cross-validation, we also found that setting the bandwidth to 1.0 works well across the datasets we considered.
>
> [1] Makoto Yamada, Denny Wu, Yao-Hung Hubert Tsai, Ichiro Takeuchi, Ruslan Salakhutdinov, Kenji Fukumizu, 2018. Post Selection Inference with Incomplete Maximum Mean Discrepancy Estimator, https://arxiv.org/pdf/1802.06226.pdf

---

### Official Review · AnonReviewer1 · 2018-11-02
**Multimodal Joint Generative Discriminative Factorization for disentangled representations with good performance and practical application (noise robustness)**

**Rating:** 7
**Confidence:** 3

**Review:**

This paper presents 'Multimodal Factorization model' that factorizes representations into shared multimodal discriminative factors and modality specific generative factors. This work applies 'Wassertein Auto-Encoders' by Tolstikhin et al (with proofs that this setup works in the multimodal case) for handling factorized joint distributions over the multimodal space. Can this method be considered as a generalization of the wasserstein autoencoder based method with a broader application? - the authors should discuss this more broadly in the paper.

Pros:
- There has been many recent work in the area of disentangling joint representations for improving generative auto-encoding architectures using VAEs, GANs, WAE and some variants of these. This work falls in this category with many interesting experiments showing SOTA generation and discrimination results on several tasks.
- This work is practical due to its robustness to noisy and/or missing data for one or more of the modalities in a multimodal machine learning classification (or generation) problem. Application of this technique for continuous multimodal time series data modeling and prediction for high accuracy requirement applications is very promising.
- The methods seems to be easily portable to other tasks. The authors say that they will make the code available to other researchers.

Cons:
- Some more comparison to other disentangling approaches such as beta-VAE, InfoGAN and partitioned VAE methods would have been useful for understanding the advantages and disadvantages of this techniques. (The authors do add a note about comparison with partitioned VAE method in the Appendix)
- For generation and classification tasks, the authors have chosen the tasks for digit recognition and sentiment analysis - I wonder if the results would hold for other types of multimodal tasks.

Overall the paper is very well-written with many experiments to support the claims.

---

> ### Author Response · Authors · 2018-11-15
> **thank you for your feedback!**
>
> Thank you for your positive comments and suggestions for improvement. We address your comments and questions below.
>
> [Regarding generalization of the Wasserstein autoencoder] Inspired by Wasserstein autoencoders, we generalize the definition of Wasserstein distance from marginal to joint distributions. This generalization enables us to perform structured prediction for multimodal learning. We believe the generalization can not only be used in multimodal learning but can also be applied in other domains. Will include this discussion in the revised manuscript.
>
> [Comparisons to additional baselines] Thank you for suggesting these additional baselines. For the beta-VAE model, we set the choice of the prior matching discrepancy as the KL-divergence and set beta large enough to encourage disentanglement of the latent variables. We train a beta-VAE to model multimodal data using the same factorization as proposed in our model (i.e. modality-specific generative factors and a multimodal discriminative factor). To provide a fair comparison to our discriminative model, we fine tune by training a classifier on top of the multimodal discriminative factor Z_y to the label. We perform experiments on 4 multimodal datasets. These experimental results have been added to section I of the appendix. MFM outperforms beta-VAE across these datasets and metrics:
> - 9.5% improvement on CMU-MOSI
> - 25.1% improvement on ICT-MMMO
> - 14.1% improvement on YouTube
> - 35.9% improvement on MOUD
>
> We have also provided additional experimental results that compare our model with partitioned VAE (Hsu and Glass, 2018) in section H of the appendix. In summary, we obtain:
> - 3.9% improvement on CMU-MOSI
> - 4.9% improvement on ICT-MMMO
> - 3.1% improvement on YouTube
> - 24.4% improvement on MOUD
>
> [Other multimodal tasks] We obtain improved results for 3 types of multimodal tasks: multimodal sentiment analysis (CMU-MOSI, ICT-MMMO, YouTube, MOUD datasets), emotion recognition (IEMOCAP dataset) and personality traits prediction (POM dataset). These results are presented in Table 1 (see page 5). Our multimodal factorization model can also be combined with many of the existing discriminative models. In future work we will explore applications of multimodal factorization to various additional tasks, such as Video QA, Image retrieval, etc. We will also release the code so that our model can be adapted to other multimodal tasks.

---

> > ### Comment · AnonReviewer1 · 2018-12-06
> > **thanks**
> >
> > Thanks for addressing some of my feedback.

---

### Meta-Review · Area_Chair1 · 2018-12-15
**Novel perspective for learning latent multimodal representations**

**Confidence:** 4
**Recommendation:** Accept (Poster)

**Metareview:**

This paper offers a novel perspective for learning latent multimodal representations. The idea of segmenting the information into multimodal discriminative and modality-specific generating factors is found to be intriguing by all reviewers and the AC. The technical derivations allow for an efficient implementation of this idea.

There have been some concerns regarding the experimental section, but they have all been addressed adequately during the rebuttal period. Therefore the AC suggests this paper for acceptance. It is an overall nice and well-thought work.